# Do workers accumulate resources during continuous employment and lose them during unemployment, and what does that mean for their subjective well-being?

Maria K. Pavlova[1,2]*

1 Institute of Psychology, Faculty of Social and Behavioral Sciences, University of Jena, Jena, Germany,
2 Institute of Gerontology, Faculty of Educational and Social Sciences, University of Vechta, Vechta, Germany

* maria.pavlova@uni-vechta.de

## Abstract

Drawing on cumulative advantage/disadvantage and conservation of resources theories, I investigated changes in economic, social, and personal resources and in subjective well-being (SWB) of workers as they stayed continuously employed or continuously unemployed. I considered age, gender, and SES as potential amplifiers of inequality in resources and SWB. Using 28 yearly waves from the German Socio-Economic Panel (SOEP 1985–2012), I conducted multilevel analysis with observations nested within participants. A longer duration of continuous employment predicted slightly higher economic resources and thereby slightly higher SWB over time. A longer organizational tenure had mixed effects on resources and predicted slight reductions in SWB via lower mastery. A longer duration of continuous unemployment predicted marked reductions mainly in economic but also in social resources, which led to modest SWB decreases. Younger workers, women, and workers with higher SES benefited from longer continuous employment and organizational tenure more. At the between-person level, some evidence for self-selection of less resourceful individuals into long-term or repeated unemployment emerged. The highly regulated German labor market and social security system may both dampen the rewards of a strong labor force attachment and buffer against the losses of long-term unemployment.

## Introduction

Long-term unemployment is seen as a "destructive and chronic social issue" [1], because long-term unemployed workers not only have decreasing chances in the labor market but also experience financial difficulties, social strain, and mental health problems. This statement is unsurprising. But consider the following one: Continuous, long-term employment is a constructive and sustainable social solution, because continuously employed workers improve their chances in the labor market and experience financial, social, and mental health benefits. Do you agree? Or unsure why this should be the case?

**Data Availability Statement:** The data underlying the results presented in the study are available from the DIW Berlin via a data distribution contract. The dataset doi: https://doi.org/10.5684/soep.v30.

**Funding:** The author received no specific funding for this work.

**Competing interests:** The author has declared that no competing interests exist.

The adverse effects of unemployment on mental health are well established [2–4], and there is a broad theoretical consensus that these effects have to do with a loss of something that is inherent to (good) employment [5–8]. Many empirical studies have supported this notion, showing that the adverse effects of unemployment can be attributed to a loss of various economic and psychosocial resources [9–14]. Likewise, there is a consensus that long-term unemployment is worse than a short-term one [3, 4, 15–17]. Is it because the long-term unemployed lose more of the same resources (e.g., social support deterioration), or because they face additional challenges (e.g., stigmatizing), or because their mental health issues or poor resources have set them at risk for (long-term) unemployment in the first place? While evidence suggests that all of this may apply to some extent [4, 15, 17–23], there is actually not much solid longitudinal research on the first possibility (i.e., resource depletion). This is remarkable, given the firmly established belief that unemployment deprives workers of essential resources associated with employment.

Even more remarkable is that researchers have given little thought to what happens to these resources if workers stay employed. It would make a practical and theoretical difference, though, whether continuously employed workers derive benefits from this circumstance (after all, this would be an additional incentive to stay continuously employed; cf. [24]) or all reward these individuals receive is not losing what they have. This perspective opens up a host of additional questions, such as who is likelier to gain resources during continuous employment, which kind of resources may be accumulated, whether staying with the same employer or changing employers is more beneficial in this regard, and whether, as a counterpart to what we assume for long-term unemployment, continuous employment leads to positive mental health outcomes? Indeed, several studies found that this might be the case [4, 25–29].

In the present paper, I draw on two theoretical perspectives that allow for considering long-term employment and unemployment as two sides of the same coin, one reflecting a stable position of advantage and another that of disadvantage (cf. [23]): life-course cumulative (dis) advantage theory [30–34] and conservation of resources theory [35, 36]. Using large-scale longitudinal data from the German Socio-Economic Panel (1985–2012), I tested whether continuous employment would lead to resource accumulation and thereby to mental health improvements over time (operationalized via subjective well-being, SWB; [37]), whereas continuous unemployment would have the opposite effects. I considered multiple resource domains (economic, social, and personal); differentiated between continuous employment and organizational tenure; and tested whether workers of different age, gender, and socioeconomic status (SES) would experience resource accumulation or depletion at a different pace.

Germany is an interesting context to address my research questions, because both stable employment and long-term unemployment were highly prevalent during the observation period. In Germany, starting with the mid-1980s, a sequence of reforms led to increasing labor market dualization into the core and marginal labor force and its flexibilization [38]. To survive increasing competition, the core labor force had to become more flexible in their wage requirements and working times, but they preserved a strong dismissal protection. Job mobility in the core labor force remained relatively low: In the SOEP, the yearly rate of external job transitions in high-skilled German employees was 6% in 1984 and 8% in 2011, reaching up to 12% during the periods of economic growth [39]. Moreover, unemployment protection was very strong up until the mid-2000s, which contributed to high rates of long-term unemployment [38]. In this context of high stability and labor market dualization, the issue of rising inequality gains in importance. A rather traditional division of gender roles, with many women experiencing prolonged career interruptions or working part-time, and the rigid system of educational qualifications, which hinders occupational mobility, make gender and SES

potentially salient amplifiers of inequality [40]. Admittedly, the generous German social security system may buffer the economic impact of unemployment.

## Unemployment, its duration, and their consequences

**Main effects.** Two meta-analyses concluded that unemployment is associated with poorer mental health [3, 4]. Its widely acknowledged theoretical explanations [8] refer to two kinds of deprivation associated with unemployment: economic (i.e., loss of income [5]) and psychological (e.g., loss of time structure, social contact, collective purpose, status, and activity [6]; lack of "vitamins", such as opportunity for personal control [7]). Cross-sectional and longitudinal studies from Australia and Europe have supported both perspectives [8]. That is, variables such as income loss or lacking social contact significantly differentiated between employed and unemployed individuals and, if tested, mediated the negative effect of unemployment on mental health [9–14, 41].

Although there is evidence that longer-term unemployment is associated with worse mental health [3, 16, 17], many longitudinal studies showed only very small mental health deterioration among continuously unemployed individuals over time [4, 15]. This inconsistency in findings might be due to self-selection of individuals with worse mental health into long-term unemployment [4, 15, 19, 22], but also to longitudinal studies failing to differentiate between job loss as an event and unemployment as a stable status. Research on life events [2, 42–45] has described the emotional reactions of adaptation (i.e., the immediate negative reaction to the unemployment event is typically followed by a partial recovery in well-being), anticipation (i.e., apprehending job loss or hoping to get a job prior to the event itself), and sensitization (i.e., the reaction to each subsequent unemployment event is increasingly negative). Such reactions may mask the long-term, cumulative effects of unemployment duration; for instance, a partial recovery in well-being during the first weeks after job loss leads to positive effects of unemployment duration on well-being in the short term.

Furthermore, a handful of mostly cross-sectional studies have addressed the associations between unemployment duration and outcomes other than mental health and found indications of resource loss across multiple domains. Specifically, the longer-term unemployed reported lower economic resources [20, 23] and higher financial worries [17], higher loneliness and fewer social contacts [18], lower perceived social support [23], greater passivity and disorientation [20], and lower self-confidence [18]. However, to my knowledge, no prior longitudinal study has tested whether resource depletion in multiple domains accounts for the presumably negative effects of long-term unemployment on mental health.

**Sociodemographic moderators.** Variation in the effects of unemployment duration along the major sociodemographic dimensions of age, gender, and SES has not received sufficient attention either. Some findings suggest that unemployment itself has stronger negative effects on mental health of younger in comparison to older workers [2–4, 15]. However, age differences in the effects of unemployment duration were only investigated in two early cross-sectional studies with men samples, which found such effects to be somewhat stronger in middle-aged workers [46, 47]. Furthermore, two meta-analyses delivered conflicting evidence on the role of gender (stronger effects of unemployment in men [4]; no gender differences [2]). Men do appear to be more negatively affected by unemployment in societies with a more traditional gender division of labor [18, 42, 44, 48]. In contrast, very few studies examined gender differences in the effects of unemployment duration, and they found no substantial differences [18, 49]. Concerning SES, the meta-analyses found that better educated individuals showed better mental health during unemployment [3] and that the adverse effect of unemployment on mental health was somewhat larger in blue-collar than in white-collar samples [4]. However, hardly any studies addressed SES differences in the impact of unemployment duration.

## Continuous employment, organizational tenure, and mental health

**Main effects.**   Cross-sectional and longitudinal research from Australia, USA, and Europe found more continuous employment to be associated with more positive (trajectories of) mental health [4, 25–29] and locus of control [28]. These findings suggest that continuous employment might protect mental health, even though the mechanisms behind this effect have hardly been explored. However, prior research has mostly used highly selective samples, employed a very limited number of measurements, or compared individuals with different employment trajectories rather than addressing within-individual change over time.

In a cross-sectional study, Pavlova and Silbereisen [23] used organizational tenure as a proxy for continuous employment and found positive effects on perceived occupational security and income, and via these on SWB. However, staying with the same firm is different from staying continuously employed but changing employers (i.e., external job mobility). On the one hand, organizational tenure reflects accumulated firm-specific work experience and presumably the degree of commitment to the firm, which may both improve performance and raise the likelihood of salary increases and promotions [50, 51]. In their meta-analysis, Ng and Feldman [50] found that organizational tenure was related to salary and promotions both directly and indirectly, via increased conscientiousness and job performance. This evidence is in favor of resource and skill accumulation during one's time with the firm, although the consequences for mental health remain unclear. On the other hand, voluntary external job mobility reportedly has positive effects on salary gains, job satisfaction, and mental health [52, 53], findings that call into doubt the benefits of long organizational tenure. However, the effects of job mobility on SWB seem to be short term [52], suggesting a quick adaptation of SWB to job change (cf. [2]). Thus, it remains unclear whether a longer organizational tenure promotes accumulation of resources and mental health.

**Sociodemographic moderators.**   Many prior studies have assumed, rather than tested, that continuous employment is more important in early careers [23, 25, 27–29]. In their meta-analysis, Ng and Feldman [50] found that age did not moderate the link between organizational tenure and salary or promotions, but it is unclear whether this finding would apply to other outcomes. Concerning gender, women experience employment interruptions for family reasons more often than men do, but such interruptions appear to have less negative effects on subsequent wages in women [54]. By implication, continuous employment might have stronger positive effects in men than in women. Similarly, the link between organizational tenure and salary was found to be weaker in women [55]. Outcomes other than salary have not been investigated. Concerning the role of SES, one study found the wage penalty for employment interruptions to be larger in lower-skilled workers, presumably because their skills and knowledge are more specific and become obsolete faster [56].

## Theoretical framework

**Cumulative advantage and disadvantage theory.**   Cumulative advantage and disadvantage (CAD) theory is a family of sociological models that describe and explain the increasing interindividual inequality in resources, functioning, and well-being with the passage of time [30, 33, 34]; cf. cumulative inequality theory [32, 57]. The key idea is that growth of inequality can be due to systemic tendencies, such as path dependency (i.e., past experiences persist into the future), which are blind to individual merit and consequently unfair [30, 33]. Several specific mechanisms may drive increasing inequality [30, 33, 34]: a higher rate of return on larger initial resources ("the rich get richer"), duration of exposure to a favorable or unfavorable condition (e.g., wealth vs. poverty), and sequential contingency of statuses (e.g., early educational attainment predicts later occupational status and earnings). Individuals exposed to initial and/

or prolonged advantage have an opportunity to accumulate desirable objects (which can be anything from money to physical fitness). Moreover, such accumulation may spread across several domains (e.g., *both* money and physical fitness; [32]). In contrast, individuals exposed to disadvantage are at risk of accumulating deficits in one or many domains. Examining multiple outcome domains is important to understand the scope of cumulative (dis)advantage [32].

As shown in Fig 1, I use the CAD concept of *duration of exposure* to understand the consequences of continuous employment and unemployment (left part of Fig 1). Cumulative effects should be evinced by a dose–response relationship between employment status duration and outcomes [32]: Continuously employed individuals will accumulate resources the longer they stay employed, whereas the opposite will happen during unemployment. Indeed, the psychological deprivation perspective on unemployment suggests that not only economic but also psychosocial resources may be affected by long-term (un)employment [6, 8]. Although the

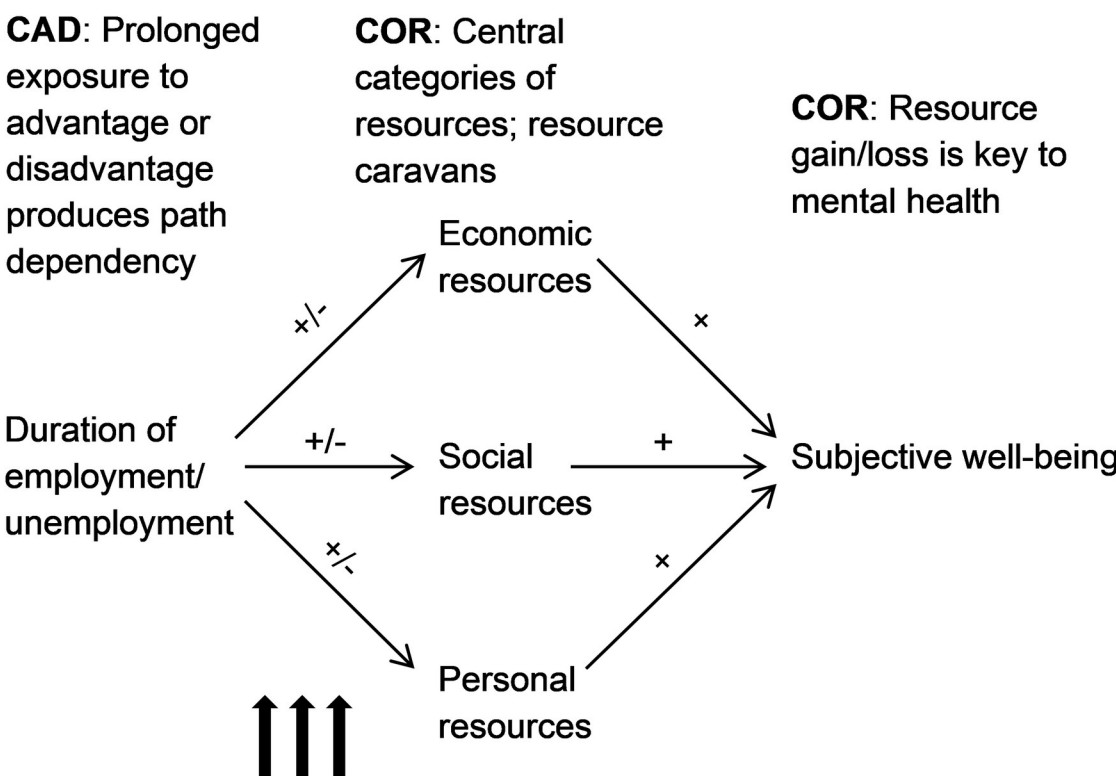

**Fig 1. Conceptual model.** CAD = cumulative (dis)advantage theory. COR = conservation of resources theory. The sign before slash refers to the effects of continuous employment; the sign after slash refers to the effects of continuous unemployment.

literature reviewed above generally supports these ideas (for long-term unemployment: [17, 18, 20, 23]; for stable employment: [23, 28]), a comprehensive, methodologically rigorous test of resource accumulation and depletion across multiple domains is still outstanding.

Furthermore, employment status cannot be the only source of inequality, and inequality is unlikely to grow in a linear fashion. First, CAD theory underscores the importance of timing of exposure to advantage or disadvantage; typically, an earlier timing (during the so-called "sensitive period") leads to a larger cumulative (dis)advantage [32, 34]. In a competitive environment, early career stages may be decisive in achieving a first advantage in skills, experience, and resources [30]. However, to my knowledge, only dated literature on the moderating role of age in the effects of long-term unemployment [46, 47] and no such empirical test for stable employment are available.

Second, an equal pace of resource accumulation for all continuously employed workers cannot be expected, because working conditions such as demands and control vary substantially across jobs [58]. The pyramid structure of organizations implies that some workers will move up the career ladder, whereas others will not [30, 31]. Along with race, which I do not consider in the present study because my sample is racially homogeneous, gender and SES have featured as major sources of inequality, particularly in the labor market [30, 34].

Hours worked and salaries of men and women have been converging over the last decades, but the gender gaps persist internationally, which may be attributed to different social norms and preferences of men and women, different occupational choices, and gender discrimination [49, 59]. A lower labor force attachment may dampen some women's ability to accumulate resources during continuous employment, but it may also buffer them against resource loss during continuous unemployment (e.g., because other normative roles are available or because some women rely on their partners as primary breadwinners; [4, 49]). Although the few studies that tested whether the effects of long-term unemployment differed between men and women did not find evidence for such differences [18, 49], the role of gender may be more pronounced in countries with a more traditional division of labor [18, 42, 44, 48]. For continuous employment or organizational tenure, evidence is available for its weaker connection to women's wages [54, 55].

Regarding SES, higher education and occupational status boost careers and facilitate positive work experiences [50, 60], which may promote resource accumulation during continuous employment. As a higher SES is related to higher coping resources, such as self-esteem, control beliefs, and expectations of finding a job [3, 4], it may also buffer against the negative effects of prolonged unemployment. However, I am not aware of any empirical research that would provide direct evidence for or against these propositions.

## Conservation of resources theory

The CAD theory is unspecific with regard to which (un)desirable objects can be accumulated. In the work context in particular, not all of the presumable benefits of employment [5–7] can be conceptualized as the objects of accumulation or depletion. For instance, time structure or regular activity will hardly show any substantial increment during stable employment or systematic deterioration during stable unemployment. I therefore turn to the conservation of resources (COR) theory [35, 36], which provides additional insight into the categories of resources, their dynamics, and their relation to mental health (see the middle and right part of Fig 1). Hobfoll [35] argues that individuals seek to obtain, retain, and protect resources and that stress occurs when resources are lost or threatened or when one fails to gain resources after a considerable resource investment. Along with material resources, whose dynamics is thought to be central in evoking stress or protecting against it, COR theory acknowledges the

importance of key personal (e.g., control beliefs and optimism) and social (e.g., social support) resources to mental health [35]. The three categories of material, social, and personal resources overlap with some of the presumable benefits of employment, such as income [5], social contact and social support [6, 7], and opportunity for personal control [7]. Whereas social resources are seen as changeable, key personal resources have been conceptualized as trait-like, highly stable constructs that enable effective management of other resources, including career-specific ones [35, 61]. However, recent evidence on personality change [62] and on some individuals' experiencing declines in perceived control after unemployment [63] suggests that a change in key personal resources is also possible.

Similarly to CAD, COR theory posits that individuals typically gain and possess (or lack) clusters of resources (resource caravans; [35, 36]). Both theories concur in describing resource gain spirals or cumulative advantage as being more difficult to launch, more incremental, and less path dependent than resource loss or cumulative disadvantage, because resource gain requires investment of existing resources [35, 36] and because advantaged conditions permit more individual choices, leading to a greater heterogeneity [34].

## Methodological challenges

Much research on the effects of unemployment duration, employment duration, or organizational tenure has been cross-sectional. Moreover, even longitudinal research that compares continuously employed and continuously unemployed individuals over time (the most common design) cannot differentiate between intraindividual change (e.g., whether workers experience change in outcomes the longer they stay employed or unemployed) and interindividual heterogeneity (e.g., the differences between individuals that make them more or less prone to stay (un)employed; [64]). However, selection effects should be taken into account in research on (un)employment, because poor mental health and low psychosocial resources are risk factors for losing a job and staying unemployed [4, 15, 19, 22]. To date, only research on reactions to life events has used a multilevel design, which separates within- and between-person variance. At the within level, intraindividual change is analyzed (e.g., how occasion-specific scores on SWB change in response to an unemployment event), while at the between level, interindividual differences (e.g., in the frequency of unemployment and in average SWB) are taken into account [42–45]. Such methodology adjusts for selection effects better than traditional longitudinal designs do [64, 65], but it requires a substantial number of repeated measurements per person. I am not aware of any prior studies that have applied a multilevel design to investigate the effects of long-term unemployment or stable employment.

## The present study

This study makes a threefold contribution to the literature. Conceptually, it applies the CAD and COR theories to juxtapose resource accumulation and loss during continuous employment and unemployment. Empirically, it investigates the pathways hitherto insufficiently explored, such as the effects of employment duration on various resources, the mediating roles of resources in the effects of (un)employment duration on SWB, and the moderating roles of age, gender, and SES. Methodologically, it utilizes a multilevel longitudinal design that disentangles the effects of (un)employment duration on resources and SWB within persons over time from interindividual differences.

The data come from 28 annual waves of the German Socio-Economic Panel (SOEP), from 1985 to 2012. As predictors, I considered the duration of continuous employment, organizational tenure, and continuous unemployment. As central outcomes, I used cognitive and emotional SWB [37]. To investigate the spread of (dis)advantage across multiple domains, I

identified potential mediator variables that belonged to one of the three resource categories described by COR theory, had featured in the literature on (un)employment and mental health, and were assessed in both working and nonworking SOEP participants at multiple waves.

Drawing on my conceptual model (Fig 1), I hypothesized that increases in the duration of both continuous employment and organizational tenure would predict a growth in economic, social, and personal resources over time (Hypothesis 1a). (I addressed potential differences in the effects of continuous employment and organizational tenure in an exploratory fashion.) In contrast, a longer duration of continuous unemployment was expected to predict decreases in these three categories of resources (Hypothesis 1b). As regards the moderators, I hypothesized that the above effects of employment status duration, positive or negative, would be stronger in younger than in older workers (Hypothesis 2) and in men than in women (Hypothesis 3). In contrast, I expected a higher SES to magnify the positive effects of continuous employment and organizational tenure on resources (Hypothesis 4a) and to buffer the negative effects of continuous unemployment on resources (Hypothesis 4b). Finally, I expected the growth or decline in economic, social, and personal resources to mediate the link between employment status duration and SWB changes (Hypothesis 5).

## Materials and methods

### Participants and procedure

The SOEP [66] started in 1984 in the FRG as a representative survey of private households, which have been followed up yearly. In the present study, I used six SOEP samples: two original West and East German samples (A and C) and the refreshments from 1998 (E), 2000 (F), 2006 (H), and 2011 (J). I did not use parts of the SOEP that oversampled special populations, such as migrants or those with high income. In the main SOEP samples, retention rates at each subsequent wave were very good (89.3–100.0% of the sample size at the previous wave; [67]). Along with ignorable causes such as death, refusals were the major cause of dropout. For each participant, I considered only those yearly observations when the person was employed or unemployed. Consequently, individuals who were out of the labor market (e.g., retired) at all observations were excluded from the study sample. Table 1 shows the number of records in the dataset for each subsample in the analytical sample.

### Measures

**Employment status.** At each wave, I coded participants as employed if they reported performing any paid work or being in vocational training or apprenticeship. University students were not considered as employed unless they reported working for pay. For unemployment, I used the ILO definition: not working, actively looking for a job, and able to start a new position within a few weeks. The corresponding items have been administered since 1994. For earlier waves, I used expected timing of (re-)entering paid employment ("as soon as possible") and immediate availability for a new position ("yes") as proxies for ILO unemployment criteria.

**Duration variables.** At each wave, duration of continuous employment, organizational tenure, and duration of continuous unemployment were measured in years (with two decimals). Using SPSS macro facility, I combined information from three sources to compute these duration variables: biography questionnaire (i.e., employment history in each year prior to entering the panel), yearly interviews (i.e., current employment status at each measurement occasion), and yearly calendar data (i.e., employment status in each month in the preceding calendar year). The correlation between employment and unemployment duration was close to zero at both between and within levels (S1 Table), possibly because most nonworking spells

**Table 1. Descriptive statistics for the central study variables.**

| Variable | Number of valid cases[a] | | Summary statistics[b] | |
|---|---|---|---|---|
| | Persons | Observations | M (SD) | % |
| Subsample | | | | |
| A (1984) | 14,352 | 127,337 | – | – |
| C (1990) | 6,918 | 68,943 | – | – |
| E (1998) | 2,420 | 16,376 | – | – |
| F (2000) | 13,432 | 81,367 | – | – |
| H (2006) | 2920 | 12,239 | – | – |
| J (2011) | 5,491 | 12,231 | – | – |
| Employed full time | 33,412 | 242,312 | – | 72.1 |
| Employed part time | 33,412 | 242,312 | – | 11.4 |
| Marginally employed | 33,412 | 242,312 | – | 11.2 |
| Unemployed ILO | 33,412 | 242,312 | – | 5.4 |
| Employment duration with up to 3 months interruptions[c] (0.0–68.0) | 28,105 | 200,924 | 14.2 (12.3) | – |
| Organizational tenure[c] (0.0–63.0) | 28,088 | 207,570 | 9.8 (9.7) | – |
| ILO unemployment duration[d] (0.0–21.0) | 5,139 | 10,569 | 1.1 (1.7) | – |
| Woman | 45,533 | – | – | 51.7 |
| Age in years (14.0–94.0) | 34,631 | 247,678 | 40.3 (12.9) | |
| Years of education (7.0–18.0) | 32,676 | 239,032 | 12.2 (2.6) | |
| Occupational autonomy (0–5) | 30,650 | 214,266 | 2.6 (1.2) | |
| Life satisfaction (0–10) | 33,605 | 243,922 | 7.0 (1.7) | – |
| Emotional well-being (1–5) | 17,383 | 62,227 | 3.6 (0.7) | – |
| Equivalized disposable income, in Euro (0.0–42,000.0) | 32,949 | 229,392 | 1,529.0 (857.9) | – |
| Financial worries (1–3) | 33,903 | 244,668 | 2.0 (0.7) | – |
| Perceived employability (1–3) | 31,248 | 204,475 | 2.0 (0.6) | – |
| Frequency of socializing[e] (1–20) | 30,797 | 152,213 | 10.3 (5.3) | – |
| Loneliness (1–4) | 17,134 | 45,701 | 1.7 (0.9) | – |
| Social support availability | 22,679 | 47,479 | – | 96.1 |
| Optimism[e] (1–20) | 20,629 | 81,075 | 10.7 (5.3) | – |
| Internal control beliefs[e] (1–18) | 18,830 | 48,369 | 10.6 (5.3) | – |
| External control beliefs[e] (1–20) | 18,802 | 48,287 | 9.8 (5.4) | – |

Statistics for the analysis sample are shown (i.e., with out-of-the-labor-market observations excluded). For scales such as emotional well-being, raw mean scores are shown. Dash = not applicable.

[a] Cases for which the value of the variable is known or the number of records in the dataset (for subsample sizes).

[b] Across persons and observations.

[c] For the observations when participants were employed, in years.

[d] For the observations when participants were unemployed, in years.

[e] Rescaled into quantile ranks.

did not fulfil the ILO unemployment criteria. Thus, a shorter employment duration did not imply a longer duration of unemployment experience. In contrast, the correlation between employment duration and organizational tenure was large (.6 at the within and .8 at the between level), reflecting low external job mobility of German workers. Entering both variables as predictors into the regression equations did not lead to recognizable multicollinearity problems.

*Employment duration.* For employment occasions, employment duration at a given wave was defined as the number of years the participant has been continuously working up to the

present time point. If the participant stayed continuously employed, this number of years would grow monotonically with each next wave. If the participant reported an interruption of more than 3 months, the value of employment duration was reset to zero, starting from the beginning of the next employment spell.

More specifically, for each wave $T_i$, I considered a participant to be continuously employed starting from a particular time point in the past $T_j$ if (a) according to the above criteria, the person was employed at $T_i$, (b) calendar data showed no longer than 3-month interruptions in paid employment between $T_j$ and $T_i$ (periods of vacation, vocational training, and re-training were not considered as interruptions), and (c), if $T_j$ preceded the date of entering the survey, biography data confirmed that the person was employed (or in vocational training or re-training) at each year between $T_j$ and $T_i$, with no unemployment spells. I ignored up to 3-month interruptions, because they might result from errors in the data (e.g., a longer vacation entered as a period of nonemployment) and because, even if short unemployment spells, they might represent a minor disruption in the course of continuous employment (in full models, unemployment history was controlled for). I did not calculate employment duration separately for different types of employment (i.e., full-, part-time, or marginal), because yearly calendar data on marginal employment started being collected only in 2005. I calculated employment duration as $T_i$—$T_j$, in years (with decimal places if months were known for both $T_i$ and $T_j$; same applies to all other duration variables).

*Organizational tenure.* Organizational tenure at each wave represented total time with the firm minus the total duration of employment interruptions of more than three months. If the participant remained employed and stayed with the firm, this variable would also grow monotonically with each next wave. It would grow slower if the participant stayed with the firm but experienced employment interruptions (e.g., parental leave), and it was reset to zero if the participant changed employers. For the self-employed, continuing in their current business activity was considered as organizational tenure, whereas changing activities was coded analogously to changing employers.

More specifically, for the first interview occasion when the participants reported being employed, I calculated their organizational tenure as the difference between the interview date and the self-reported date of starting working with the present company (for self-employed: starting their current business) minus the duration of any nonworking episodes in between, which I estimated from yearly calendar and biography data. For each subsequent wave $T_i$, the difference between $T_i$ and $T_{i-1}$ was added to the previous value of tenure if the participants reported no change of employer. If calendar data indicated employment interruptions of more than three months, they were subtracted from this sum. Likewise, employment interruptions of a year or more (e.g., parental leave), when the participants were not working at one or several yearly interviews and then reported to have returned to the same employer, were subtracted from the total time with the firm. If at a given interview the participants reported changing employers, I calculated organizational tenure as the difference between the interview date and the new starting date and used it as a baseline for subsequent waves until a new employer change occurred. Where the information on the starting date with the firm, employment continuity, and changes in one's employment situation was lacking or inconsistent, I double-checked all available sources of data to arrive at the most plausible estimate of organizational tenure.

*Unemployment duration.* For unemployment occasions, unemployment duration represented the number of years during which the participant remained nonworking and continued to fulfil the ILO unemployment criteria at each wave. If the nonworking participant reported a very short employment spell of even one month or not fulfilled the ILO criteria at the next interview, their duration of continuous unemployment would be reset to zero. If the

participant remained unemployed according to the ILO criteria, the value of unemployment duration would grow monotonically with each next wave.

More specifically, for each wave $T_i$, I considered a participant to be continuously unemployed (according to the ILO criteria) starting from a particular time point in the past $T_j$ if (a) the person fulfilled the ILO criteria for unemployment at $T_i$ and at each interview between $T_j$ and $T_i$, (b) calendar data showed that the person did not work (and was not in vocational training or re-training) at each month between $T_j$ and $T_i$, and (c), if $T_j$ preceded the date of entering the survey, biography data showed that the person was registered as unemployed at each year between $T_j$ and $T_i$, with no working (or vocational training) episodes. If the person fulfilled the ILO criteria at $T_i$, did not work between $T_i$ and $T_{i-1}$, but did not fulfil the ILO criteria at $T_{i-1}$, I set $T_j$ to the date (i.e., month and year) exactly between $T_i$ and $T_{i-1}$. As calendar data did not record marginal employment prior to 2005, I could not rule out that unemployed participants did have a marginal job at some point between $T_j$ and $T_i$.

**Mediators and outcomes.** *Economic resources*. I calculated equivalized disposable income by applying the modified OECD equivalence scale to net monthly household income in Euro (a generated variable available at each wave; converted from DM for earlier waves). The financial worries item was administered at each wave as part of a larger list of worries ("How concerned are you about... your own economic situation?" 1 = not concerned at all; 3 = very concerned; for previous use, see [68]). One item on perceived employability (cf. career outlook in Warr's "vitamin" model [7]) has been administered yearly since 1987 (for previous use, see [69]). It was worded slightly differently for working ("If you lost your job today, would it be easy, difficult, or almost impossible for you to find a new job that is at least as good as your current one?" 1 = almost impossible; 2 = difficult; 3 = easy) and nonworking ("If you were currently looking for a new job, would it be easy, difficult, or almost impossible to find an appropriate position?") participants. Financial worries and perceived employability correlated both with objective (e.g., age, region, and SES) and subjective (e.g., mastery) indicators of (dis) advantage, which supported their validity (S1 Table).

*Social resources*. Frequency of socializing with friends, relatives, and acquaintances (cf. social contact as a latent benefit of employment; [6]) has been measured at least at every second wave in two alternating versions. The first version includes one item: "Meeting friends, relatives, or neighbors" (1 = never; 2 = less often; 3 = every month; 4 = every week). The second version includes two items and has a slightly different response format: "Visiting or being visited by neighbors, friends, or acquaintances" and "Visiting or being visited by family members or relatives" (1 = never; 2 = seldom; 3 = at least once a month; 4 = at least once a week; 5 = daily). These items do not explicitly mention colleagues, which is an advantage because the formulation is applicable to both working and nonworking participants. Loneliness (one item; "I often feel lonely;" 1 = does not apply at all; 4 = fully applies) was measured six times with irregular intervals (mostly in the 1990s) as part of an anomie scale, which was adapted from the earlier German Welfare Survey [70]. Finally, a set of items originating from the same survey referred to participants' social networks and were administered five times with 5-year intervals. I used one item to assess social support availability in dire situations (cf. [7, 18]): "If you were in need of long-term care (for example, in the case of bad accident), who would you ask for help?" (0 = at least one person named; 1 = no one). As expected, more subjective measures (loneliness and social support availability) correlated more strongly with each other than with the more objective measure (frequency of socializing; S1 Table).

*Personal resources*. As personal resources such as sense of control, optimism, and self-esteem tend to be highly correlated [35], I decided to use optimism and control beliefs as indicators for a broader construct of mastery. I used two alternating versions of one item to assess optimism. The first version was administered six times as part of the anomie scale: "When I

think about the future, I am actually quite optimistic" (1 = does not apply at all; 4 = fully applies). The second version, developed and validated by Trommsdorff [71], was administered four times: "When you think about the future, are you. . . optimistic, more optimistic than pessimistic, more pessimistic than optimistic, pessimistic?" (1 = pessimistic; 4 = optimistic). Furthermore, two sets of items measured control beliefs after Rotter [72]. The first set was administered at earlier waves (i.e., 1994, 1995, and 1996) and substituted with another set at later waves (i.e., 1999, 2005, and 2010). I identified items from both sets that were semantically close and manifested the largest intercorrelations across time. For internal control, these were "I can control much of what happens in my life" from the earlier version and "How my life goes depends on me" from the later version, $r_{1996-1999} = .27$. For external control, these were two items from the earlier version ("I believe that nobody can avoid their fate. What is going to happen will happen" and "When I get what I want, it is mostly because of luck," $r_{1996} = .39$; I computed their mean score) and one item from the later version ("What a person achieves in life is above all a question of fate or luck," $r_{1996-1999} = .36$). In both versions, the response scale was from "do not agree at all" to "fully agree," which was a four-point scale up until 1999 and a seven-point scale in 2005 and 2010.

*Subjective well-being.* A single item on general life satisfaction has been administered at all waves ("How satisfied are you with your life, all things considered?" 0 = completely dissatisfied; 10 = completely satisfied). A four-item scale on emotional well-being has been administered yearly since 2007 (for previous use, see [45]). This scale assesses the frequency of experiencing anger, worry, happiness, and sadness in the past four weeks (1 = very rarely; 5 = very often; α = .65–.68).

**Equating alternating item versions.** Frequency of socializing, optimism, and control beliefs were assessed with alternating item versions, and no anchor items (i.e., items that would be exactly the same in both versions) were available. In such cases, there is no perfect way of equating different items (e.g., I could not use approaches based on the item response theory; [73]). To bring alternating item versions to approximately the same scale, I applied distribution-based ranking with 20 quantiles to each of these variables within each wave. The new scores showed the approximate position of a participant's value in the overall score distribution. Such distribution-based equating is analogous to but more precise than *z*-standardization if the scores are not normally distributed [73]. In subsequent analyses, I added a dummy variable for the waves when a different item version was used as a predictor in the regression equation (not shown in tables).

**Moderator variables at the between level.** Average age across observations represented the approximate timing of (un)employment spells in the individual life course. It was an acceptable proxy because most participants stayed in the survey for a limited number of years (seven on average). Gender was a binary variable. Educational attainment included all educational levels and was measured in years. Finally, occupational autonomy was assessed in employed participants and classified their occupational positions based on required years of training, number of dependent employees, expected task complexity, etc. [74] (0 = apprenticeship, internship, etc.; 1 = low autonomy; 5 = high autonomy). To create between-level indicators, I computed individual averages on educational attainment and occupational autonomy across all measurements.

**Control variables.** Because employment trajectories have been much more erratic in the former East Germany as compared to the West, I controlled for the region of origin (i.e., living in the former GDR in 1989; 0 = no; 1 = yes; missing for those born after 1991). Basic demographic indicators included age and gender. Among time-varying covariates, I considered educational attainment (note that spells of being in education = out of the labor market were not included into the analysis) and, for employment occasions, information on regular part-time

employment of less than 30 hours per week (0 = no; 1 = yes), marginal employment (i.e., irregular or low-wage job; 0 = no; 1 = yes), fixed-term employment contract (0 = no; 1 = yes), self-employment (0 = no; 1 = yes), occupational autonomy, and working overtime in hours (0 = no overtime hours). To take the adaptation, sensitization, and anticipation effects into account [2, 42–45], I controlled for being in the first 6 or the last 12 months of the (un)employment spell, the current spell number, as well as the number and total duration of prior unemployment spells (in years). Finally, I included satisfaction with health (one item; 0 = completely dissatisfied; 10 = completely satisfied) and self-reported disability (0 = no; 1 = yes) to account for the fact that individuals with poorer health have less stable employment trajectories. All time-varying control variables were assessed yearly. Most of them (except for the characteristics of the current spell, which were purely within-level variables) were also included as control variables at the between level (see S2 Table for a full list of control variables at both levels).

**Analytical approach.** I conducted all analyses in MPlus (Versions 7.4 and 8.2 for Linux; [75]). The variance in both predictor and outcome variables was decomposed into within and between components [76]. At the within level, this was achieved via person-mean centering of continuous variables [65]. I included age and age squared as additional predictors (i.e., time factors) to detrend the duration variables [65]. That is, I tested the effects of employment status duration net of any time-related effects caused by other unobserved factors. The regression coefficients of duration variables showed whether a growth in duration was associated with a linear trend in outcomes over time. The effects of employment duration and organizational tenure referred to what was happening during employment occasions, whereas the effects of unemployment duration referred to unemployment occasions. For instance, if employment duration had a significantly positive effect on SWB, this would mean that, on average, SWB increased with each subsequent year of continuous employment. This approach was akin to growth curve modeling in a multilevel framework [77], excepting that the time factors (i.e., duration variables) did not always monotonically increase with each subsequent observation (i.e., after an employment interruption, employment duration was reset to zero).

At the between level, most variables represented individual averages (or rates in case of categorical variables) across all employment and unemployment occasions. In some cases, I opted for substantively more meaningful indicators, such as the total number of unemployment spells experienced during the observation period. All continuous between-level variables for which zero was not a meaningful value were grand-mean centered.

To reduce the number of variables, I modelled emotional well-being and mastery as latent variables at both within and between levels. Emotional well-being was measured by four manifest indicators: happiness, anger, sadness, and worry. Mastery represented personal resources, measured by three manifest indicators: optimism and internal and external control beliefs. The model fit was acceptable: for the two latent constructs in one model, $\chi^2$ (42, $N$ = 122,438) = 1274.5, $p$ < .001, CFI = .953, RMSEA = .015, SRMR$_{within}$ = .015, SRMR$_{between}$ = .076.

For continuous mediators or outcomes (income, frequency of socializing, mastery, life satisfaction, and emotional well-being), I conducted twolevel linear regression analyses with robust maximum likelihood (ML) estimation, which corrected for potential violations of normality assumptions and accounted for the clustering of observations not only within individuals but also within households. Missing data were handled with the full information maximum likelihood (FIML) estimator, whereby the data are not imputed, but all available information from each case is used to calculate individual likelihood functions [78]. I included the likely predictors of dropout as missing data covariates: at the within level, satisfaction with health, education, employment status, income, and dependent variable (DV; e.g., life satisfaction) from the previous wave; at the between level, individual averages on education and income across all observations.

For categorical outcomes (ordinal indicators of financial worries, perceived employability, and loneliness; a binary indicator of social support availability), I conducted twolevel probit regression analyses with Bayesian estimation because ML estimation was computationally infeasible [79]. Bayesian estimation does not provide conventional *p*-values but it does provide credibility intervals. With Bayesian estimation, it was not possible to account for clustering within households. Missing data handling was similar to FIML. Probit regression with categorical outcomes is not linear, which means that, when linear regression coefficients are translated into category probabilities, different categories of DV change with unequal rate in response to linear changes in the predictor. To illustrate effect sizes for such outcomes, I selected the most informative category of each DV (e.g., reporting high financial worries) and reported probability change for this category in percentage points (pp).

To test for mediation, after having estimated the first and second paths of mediation separately, I fit full models with multiple mediators at the within-person level (1-1-1 mediation; [80]) using Bayesian estimation. Indirect effects were computed as the products of regression coefficients from the first and the second path. I tested them for significance using asymmetric Bayesian credibility intervals [81]. To avoid multiple significance testing, I considered total indirect effects via categories of mediators (economic, social, and personal resources) rather than via specific mediators.

In separate analyses, I allowed the within-level effects of employment status duration on resources to be random (i.e., to vary across persons) and included the four moderating variables as predictors of these random slopes at the between level (i.e., cross-level interactions; [76]). This procedure was only possible with Bayesian estimation. Because of the high computational load of these models, I had to limit the sample to those observations that had no missing data on the within-level predictor (i.e., employment status duration). For this reason, I present the results for the main effects and for mediation analyses, where missing values were fully estimated, separately from moderation analyses. To obtain an upper limit of the effect size for moderated mediation, I additionally tested for moderation of the total effects of employment status duration on SWB. To correct for multiple significance testing, I interpreted only those moderation effects that were significant at $p < .01$.

## Results

Table 1 shows descriptive statistics on the central variables of interest after out-of-the-labor-market observations have been excluded (see S1 Table for correlational matrices at the within and between levels). Figures cover the observation period from 1985 to 2012 as data from the first wave of the study (1984) were used only as missing data correlates for 1985. These statistics show that unemployment occasions that fulfilled the ILO unemployment criteria were rare (about 5%) but that the average duration of continuous unemployment was sizable (about 1 year). Moreover, the average duration of continuous employment and organizational tenure was quite long (about 14 and 10 years, respectively).

### Within-level effects of unemployment and repeated spells of (un) employment

Results of fully adjusted twolevel regression analyses with resources as DVs are shown in S2 Table. In this section, I briefly describe the effects of the unemployment itself and (un)employment history that were not in the focus of this study. As expected, being unemployed reduced economic resources: On unemployment occasions, participants reported much lower equivalized disposable income (a difference of ca. 360 Euro if sample average is taken as a baseline), higher financial worries (a 18.8 pp difference in the probability of belonging to the highest

category), and lower perceived employability (a 13.8 pp difference in the probability of saying that finding an appropriate position would be almost impossible). As regards social resources, only self-reported loneliness was significantly but only slightly higher on unemployment occasions (a 2.6 pp difference in the probability of belonging to the highest category). Finally, being unemployed was significantly and sizably associated with lower occasion-specific mastery (β = -0.75). (Unless specified otherwise, the effect sizes refer to the *SD* of the continuous DV at the respective level (within or between). For binary predictors, beta coefficients refer to the amount of change (in terms of *SD*) in the DV associated with the change from 0 to 1 in the predictor variable.)

After employment interruptions (i.e., during the second and subsequent employment spells as contrasted to the first in the observation period), individuals reported significantly lower equivalized disposable income (up to 198 Euro difference), lower perceived employability (a 2.2–3.5 pp difference), and lower frequency of socializing (up to 0.13 *SD* difference), but also lower financial worries (up to 4.2 pp difference) and higher mastery (up to 0.25 *SD* difference). Moreover, the second and subsequent unemployment spells (vs. the first) were associated with lower income (up to 40 Euro difference), but there were no consistent associations with other resources. A history of prior unemployment (i.e., the number and total duration of prior unemployment spells) had significant but very small effects only on economic resources, mostly in the expected negative direction.

## Within-level effects of employment status duration on resources (first stage of mediation)

The effects of duration variables on resources are summarized in Table 2, whereas Fig 2 illustrates estimated trajectories of psychosocial resources (excepting loneliness, for which no significant effects emerged) between 0 (representing individual average) and 10 subsequent years of status duration for continuous unemployment, continuous employment, and organizational tenure. In a nutshell, Hypothesis 1a was not supported as continuous employment was associated with a limited resource accumulation only in the economic domain, whereas organizational tenure had sometimes negative effects. Hypothesis 1b was partially supported as resource depletion during continuous unemployment pertained mainly to economic, partly to social resources.

If sample average (about 1,525 Euro) is taken as a baseline, equivalized disposable income decreases from about 1,164 Euro to about 811 Euro after 10 years of unemployment, while it remains stable during continuous employment and slightly increases (by about 30 Euro in 10 years) with longer organizational tenure (Fig 2A, right vertical axis). The probability of being very concerned about one's economic situation increases from 36.7% to 63.6% after 10 years of continuous unemployment, while it marginally decreases (from 17.8% to 17.0%) during continuous employment and marginally increases (from 17.8% to 18.7%) with longer organizational tenure (Fig 2A, left vertical axis). The probability of saying that finding an appropriate position would be almost impossible increases from 34.1% to 45.3% after 10 years of continuous unemployment, whereas it increases from 20.0% by about 2 pp after 10 years of continuous employment and by about 4 pp after 10 years of tenure (Fig 2A, left vertical axis).

Concerning social resources, frequency of socializing shows a negligibly small increase by 0.03 *SD* for 10 years of tenure and is unrelated to employment or unemployment duration (Fig 2B, right vertical axis). The probability of reporting no social support available in dire situations increases from 5.6% to 28.5% after 10 years of continuous unemployment, while it stays continuously low (at about 3–4%) during employment occasions, irrespective of status duration (Fig 2B, left vertical axis). Finally, mastery (personal resources) decreases by about 0.08

**Table 2. Within-level effects of employment status duration on resources (first stage of mediation).**

| DVs and predictors | *B* | 95% CI | *p* |
|---|---|---|---|
| DV: *income* | | | |
| **Employment duration** | 0.000 | [-0.001, 0.001] | .698 |
| **Organizational tenure** | 0.002 | [0.001, 0.003] | < .001 |
| **Unemployment duration** | -0.036 | [-0.046, -0.026] | < .001 |
| DV: *financial worries* | | | |
| **Employment duration** | -0.004 | [-0.006, -0.002] | < .01 |
| **Organizational tenure** | 0.004 | [0.002, 0.006] | < .01 |
| **Unemployment duration** | 0.088 | [0.056, 0.120] | < .01 |
| DV: *perceived employability* | | | |
| **Employment duration** | -0.009 | [-0.011, -0.007] | < .01 |
| **Organizational tenure** | -0.017 | [-0.019, -0.015] | < .01 |
| **Unemployment duration** | -0.040 | [-0.072, -0.009] | < .05 |
| DV: *frequency of socializing* | | | |
| **Employment duration** | -0.008 | [-0.019, 0.002] | .120 |
| **Organizational tenure** | 0.011 | [0.001, 0.020] | .025 |
| **Unemployment duration** | -0.022 | [-0.169, 0.125] | .769 |
| DV: *loneliness* | | | |
| **Employment duration** | 0.002 | [-0.003, 0.006] | *ns* |
| **Organizational tenure** | 0.001 | [-0.004, 0.005] | *ns* |
| **Unemployment duration** | 0.041 | [-0.026, 0.103] | *ns* |
| DV: *social support availability* | | | |
| **Employment duration** | 0.004 | [-0.004, 0.013] | *ns* |
| **Organizational tenure** | -0.008 | [-0.020, 0.003] | *ns* |
| **Unemployment duration** | -0.131 | [-0.252, -0.010] | < .05 |
| DV: *mastery* | | | |
| **Employment duration** | 0.000 | [-0.007, 0.007] | .949 |
| **Organizational tenure** | -0.008 | [-0.015, -0.002] | .014 |
| **Unemployment duration** | -0.055 | [-0.143, 0.033] | .218 |

DV = dependent variable. Mastery was modeled as a latent variable. All effects are adjusted for the full set of control variables. Exact *p*-values are shown where available (ML estimation).

*SD* for 10 years of tenure, whereas it stays constant during continuous employment (Fig 2C). Interestingly, continuous unemployment is associated with a much larger decrease in mastery (about 0.50 *SD* for 10 years of unemployment), which, however, is not statistically significant.

## Moderator analyses of the first stage of mediation

Table 3 shows central findings from the twolevel models with random slopes of duration variables and multiple moderators at the between level.

**Age.** Hypothesis 2, which predicted that employment status duration would have stronger effects on resources in younger workers, was not supported by findings. In fact, workers' average age across observations significantly (at $p < .01$) moderated only the effects of employment duration and organizational tenure on perceived employability. As shown in Fig 3, in younger workers (average age 25 during the observation period), the probability to report that finding an equal/appropriate position would be almost impossible decreased from about 12% to 8–9% after 10 years of continuous employment or tenure. In older workers (average age 55), this probability increased from about 37% to 43.4% after 10 years of continuous employment and

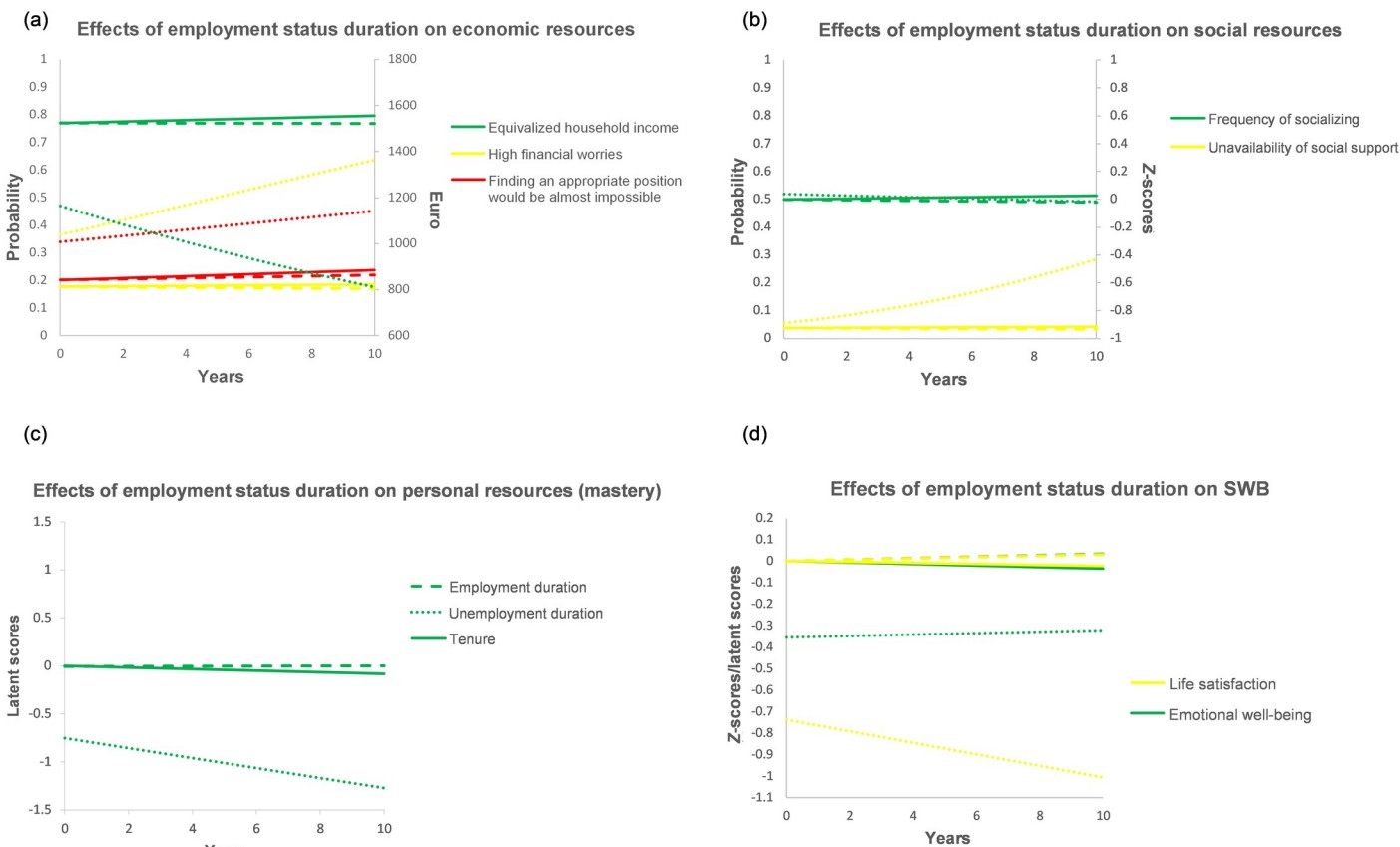

**Fig 2. Effects of employment status duration on the change in resources and SWB.** Dashed lines = effects of employment duration. Solid lines = effects of organizational tenure. Dotted lines = effects of unemployment duration. The zero point of the X-axis represents not the beginning of an (un)employment spell or of time with the firm but its person-specific mean.

to 49.1% after 10 years of organizational tenure. Thus, the effects on perceived employability were stronger and less favorable in older workers.

**Gender.** Hypothesis 3 predicted that the effects of employment status duration on resources would be stronger in men than in women. This hypothesis was supported only for the effect of unemployment duration on perceived employability, which was much more negative in men (after 10 years of unemployment, a significant increase from 33.3% to 74.6% in the probability to report that finding an equal/appropriate position would be almost impossible) than in women (a nonsignificant increase from 35.6% to 51.4%, respectively; Fig 4A). However, other significant moderating effects of gender (Table 3 and Fig 4) suggested that a longer continuous employment was generally more beneficial in women, rather than having weaker effects. First, with more years of continuous employment, women, who started out with slightly lower perceived employability, experienced a tiny improvement, which was not observed in men (Fig 4A). Second, with more years of continuous employment or organizational tenure, women reported a slight increase in the frequency of socializing (0.05 and 0.09 *SD* after 10 years, respectively), which was not observed in men either (Fig 4B). Third, men experienced a slight decrease in mastery with more years of continuous employment (0.05 *SD* after 10 years) or organizational tenure (0.10 *SD* after 10 years), whereas women experienced a slight increase in mastery with more years of continuous employment (0.07 *SD* after 10 years) and no significant change with a longer organizational tenure (Fig 4C).

**Table 3. Moderation of the within-level effects of employment status duration on resources by between-level predictors.**

| Random effects (between level) | Income (logged) | Financial worries | Perceived employability | Frequency of socializing | Loneliness | Social support availability | Mastery |
|---|---|---|---|---|---|---|---|
| Average effect of employment duration (S1) | 0.001 * | -0.010 ** | 0.000 | -0.009 | 0.005 | 0.006 | -0.009 * |
| | [0.000, 0.002] | [-0.013, -0.006] | [-0.004, 0.004] | [-0.021, 0.003] | [-0.003, 0.013] | [-0.011, 0.022] | [-0.018, 0.000] |
| S1 on average age | 0.000 | 0.000 | -0.002 ** | 0.000 | 0.001 * | 0.000 | 0.000 |
| | [0.000, 0.000] | [0.000, 0.000] | [-0.002, -0.002] | [-0.001, 0.001] | [0.000, 0.001] | [-0.001, 0.001] | [0.000, 0.001] |
| S1 on gender (woman) | 0.002 * | 0.001 | 0.007 ** | 0.024 ** | -0.008 * | -0.007 | 0.021 ** |
| | [0.001, 0.003] | [-0.003, 0.005] | [0.002, 0.012] | [0.012, 0.037] | [-0.017, 0.000] | [-0.024, 0.009] | [0.011, 0.031] |
| S1 on average years of education | 0.000 | -0.001 * | 0.005 ** | -0.004 * | -0.002 | 0.002 | 0.000 |
| | [0.000, 0.001] | [-0.002, 0.000] | [0.004, 0.006] | [-0.007, -0.001] | [-0.004, 0.000] | [-0.002, 0.006] | [-0.002, 0.003] |
| S1 on average occupational autonomy | 0.003 ** | -0.007 ** | -0.009 ** | -0.006 | 0.003 | -0.002 | -0.002 |
| | [0.002, 0.004] | [-0.009, -0.004] | [-0.012, -0.006] | [-0.014, 0.002] | [-0.002, 0.008] | [-0.013, 0.009] | [-0.009, 0.004] |
| Average effect of organizational tenure (S2) | -0.001 | 0.004 * | -0.019 ** | 0.010 | 0.001 | 0.004 | -0.017 ** |
| | [-0.002, 0.001] | [0.000, 0.008] | [-0.023, -0.014] | [-0.003, 0.022] | [-0.006, 0.009] | [-0.016, 0.025] | [-0.026, -0.007] |
| S2 on average age | 0.000 | 0.000 * | -0.002 ** | 0.000 | 0.000 | 0.000 | 0.001 * |
| | [0.000, 0.000] | [-0.001, 0.000] | [-0.002, -0.001] | [-0.001, 0.001] | [0.000, 0.001] | [-0.001, 0.001] | [0.001, 0.002] |
| S2 on gender (woman) | 0.000 | 0.000 | 0.004 | 0.025 ** | -0.003 | -0.002 | 0.022 ** |
| | [-0.001, 0.001] | [-0.004, 0.005] | [-0.002, 0.009] | [0.012, 0.039] | [-0.012, 0.005] | [-0.019, 0.016] | [0.010, 0.033] |
| S2 on average years of education | 0.000 | -0.001 | 0.004 ** | -0.002 | -0.001 | 0.001 | -0.001 |
| | [0.000, 0.001] | [-0.002, 0.000] | [0.003, 0.006] | [-0.005, 0.002] | [-0.003, 0.001] | [-0.003, 0.005] | [-0.004, 0.002] |
| S2 on average occupational autonomy | 0.002 ** | -0.007 ** | -0.004 * | -0.009 | 0.004 | -0.002 | -0.003 |
| | [0.001, 0.003] | [-0.010, -0.004] | [-0.008, -0.001] | [-0.017, 0.001] | [-0.002, 0.010] | [-0.014, 0.009] | [-0.010, 0.004] |
| Average effect of unemployment duration (S3) | -0.099 ** | 0.189 ** | -0.152 ** | 0.207 | -0.013 | 0.139 | -0.006 |
| | [-0.128, -0.073] | [0.092, 0.279] | [-0.264, -0.051] | [-0.158, 0.560] | [-0.241, 0.228] | [-0.256, 0.850] | [-0.261, 0.298] |
| S3 on average age | 0.001 | -0.003 | -0.001 | 0.005 | 0.000 | -0.009 | 0.004 |
| | [0.000, 0.002] | [-0.006, 0.001] | [-0.005, 0.003] | [-0.008, 0.016] | [-0.009, 0.011] | [-0.030, 0.006] | [-0.006, 0.015] |
| S3 on gender (woman) | 0.011 | -0.047 | 0.094 ** | 0.016 | 0.003 | 0.205 | 0.120 |
| | [-0.012, 0.032] | [-0.121, 0.018] | [0.023, 0.165] | [-0.229, 0.285] | [-0.174, 0.147] | [-0.136, 0.531] | [-0.092, 0.350] |
| S3 on average years of education | 0.001 | 0.012 | -0.005 | 0.048 | -0.013 | -0.035 | 0.035 |
| | [-0.006, 0.007] | [-0.010, 0.037] | [-0.026, 0.015] | [-0.021, 0.121] | [-0.067, 0.043] | [-0.117, 0.039] | [-0.038, 0.102] |
| S3 on average occupational autonomy | -0.028 ** | 0.042 | -0.039 | 0.089 | -0.018 | 0.113 | 0.041 |
| | [-0.043, -0.014] | [-0.015, 0.088] | [-0.102, 0.014] | [-0.117, 0.259] | [-0.135, 0.106] | [-0.067, 0.411] | [-0.087, 0.194] |
| Residual variance ($\sigma^2$) S1 | 0.001 ** | 0.005 ** | 0.007 ** | 0.019 ** | 0.005 ** | 0.009 ** | 0.007 ** |
| | [0.001, 0.001] | [0.004, 0.005] | [0.007, 0.008] | [0.017, 0.021] | [0.004, 0.005] | [0.007, 0.011] | [0.005, 0.009] |
| Residual variance ($\sigma^2$) S2 | 0.001 ** | 0.005 ** | 0.008 ** | 0.020 ** | 0.005 ** | 0.010 ** | 0.007 ** |
| | [0.001, 0.001] | [0.005, 0.006] | [0.007, 0.009] | [0.018, 0.023] | [0.005, 0.006] | [0.008, 0.012] | [0.006, 0.009] |

(*Continued*)

**Table 3.** (Continued)

| Random effects (between level) | Income (logged) | Financial worries | Perceived employability | Frequency of socializing | Loneliness | Social support availability | Mastery |
|---|---|---|---|---|---|---|---|
| Residual variance (σ²) S3 | 0.021 ** | 0.048 ** | 0.035 ** | 0.261 ** | 0.103 ** | 0.311 ** | 0.175 ** |
|  | [0.015, 0.027] | [0.034, 0.068] | [0.025, 0.052] | [0.115, 0.543] | [0.062, 0.179] | [0.108, 0.620] | [0.085, 0.441] |

Cells show unstandardized regression coefficients with Bayesian credibility intervals in square brackets. Random S1 and S3 were estimated in the same model (except for unavailability of social support and mastery, for which S1 and S3 were estimated separately because of convergence problems), random S2 was estimated in a separate model. Missing values on the predictor (employment status duration) could not be estimated in these models. For analyses with random S1 and S3, $N_{persons}$ = 28,911, $N_{observations}$ = 211,480. For analyses with random S2, $N_{persons}$ = 30,045, $N_{observations}$ = 223,312. The models were adjusted for the full set of control variables.

* $p < .05$.

** $p < .01$.

**Socioeconomic status.** Average educational attainment across observations significantly (at $p < .01$) moderated only the effects of employment duration and organizational tenure on perceived employability (Table 3). These moderation effects were in line with Hypothesis 4a: With more years of continuous employment, perceived employability slightly improved in those with higher educational attainment, whereas it slightly decreased in those with lower educational attainment (Fig 5A). Moreover, perceived employability generally worsened with longer organizational tenure, but this effect was stronger in lower educated than in higher

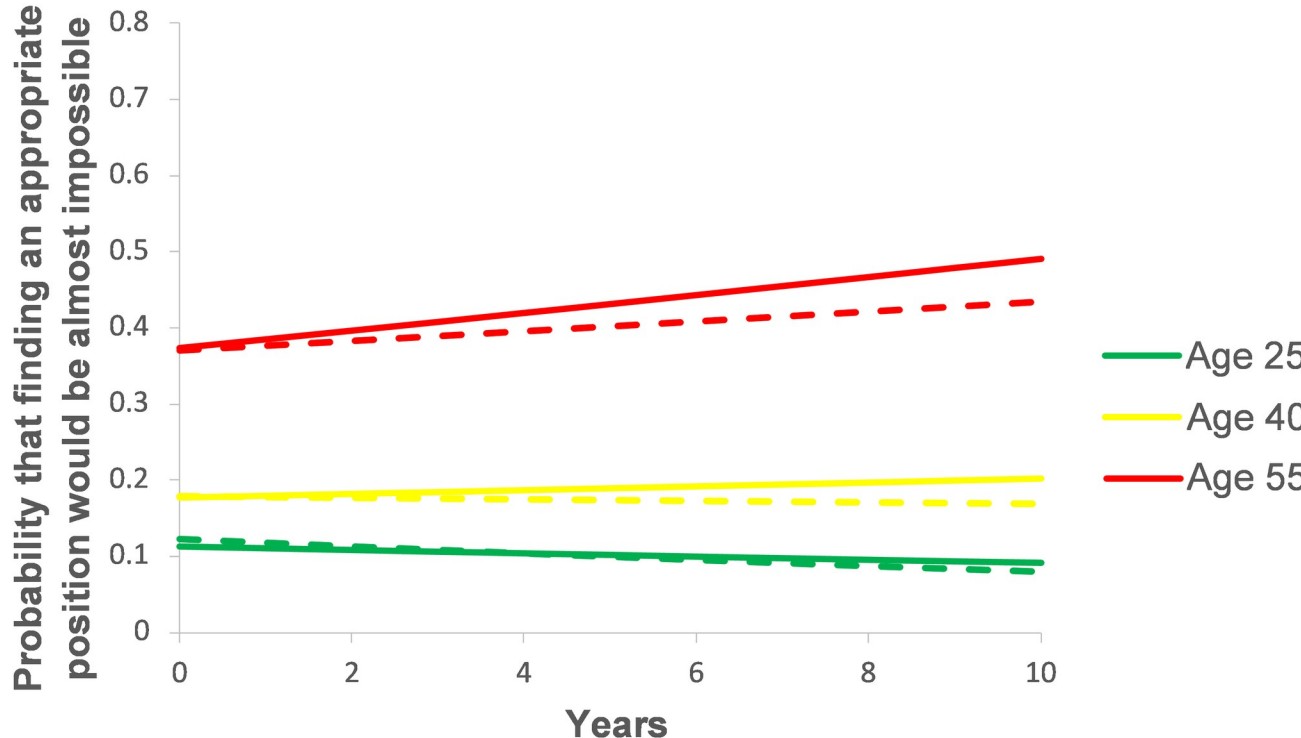

**Fig 3. Moderation of the within-level effects of employment status duration by person-specific mean age across observations.** Dashed lines = effects of employment duration. Solid lines = effects of organizational tenure.

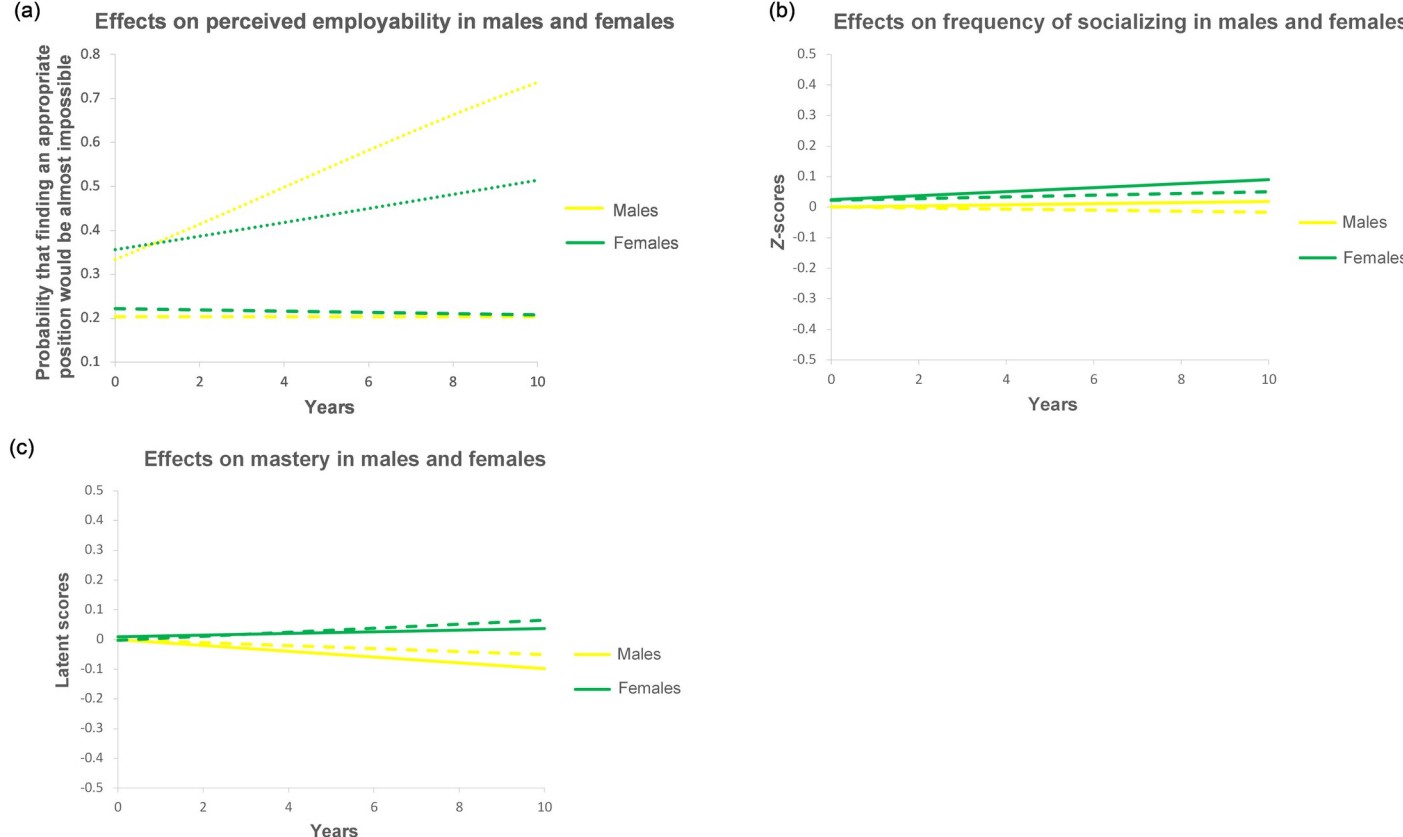

**Fig 4. Moderation of the within-level effects of employment status duration by gender.** To show potential gender differences in the average level of resources, effects on frequency of socializing and mastery are standardized on the basis of full variance of the DV (within + between). Dashed lines = effects of employment duration. Solid lines = effects of organizational tenure. Dotted lines = effects of unemployment duration.

educated workers (e.g., the probability to report that finding an equal/appropriate position would be almost impossible increased by 6.4 pp per 10 years of tenure in the former vs. 1.8 pp per 10 years in the latter). In contrast, workers with higher average occupational autonomy across observations experienced a slight deterioration in perceived employability with longer duration of continuous employment, whereas their counterparts with lower occupational autonomy experienced a slight improvement (Fig 5B; the interaction effect was significant at $p < .01$, Table 3). These effects did not support Hypothesis 4a, but they might indicate that workers with better jobs considered their position to be exceptional and therefore difficult to exchange for an equally good one.

Occupational autonomy significantly (at $p < .01$) moderated the effects of employment duration and organizational tenure on equivalized disposable income and financial worries (Table 3 and Fig 5C and 5D). In line with Hypothesis 4a, workers with higher occupational autonomy reported substantially higher income and lower financial worries than those with lower occupational autonomy, and this gap slightly increased with a longer duration of continuous employment or tenure. For instance, the estimated average gap in equivalized disposable income between those 1 *SD* below and 1 *SD* above average occupational autonomy grew from 344 Euro to 450 Euro after 10 years of continuous employment and from 361 Euro to 427 Euro after 10 years of organizational tenure. The respective difference in financial worries increased from 9.5 pp to 12.2 pp after 10 years of continuous employment and from 9.2 pp to 13.3 pp after 10 years of tenure.

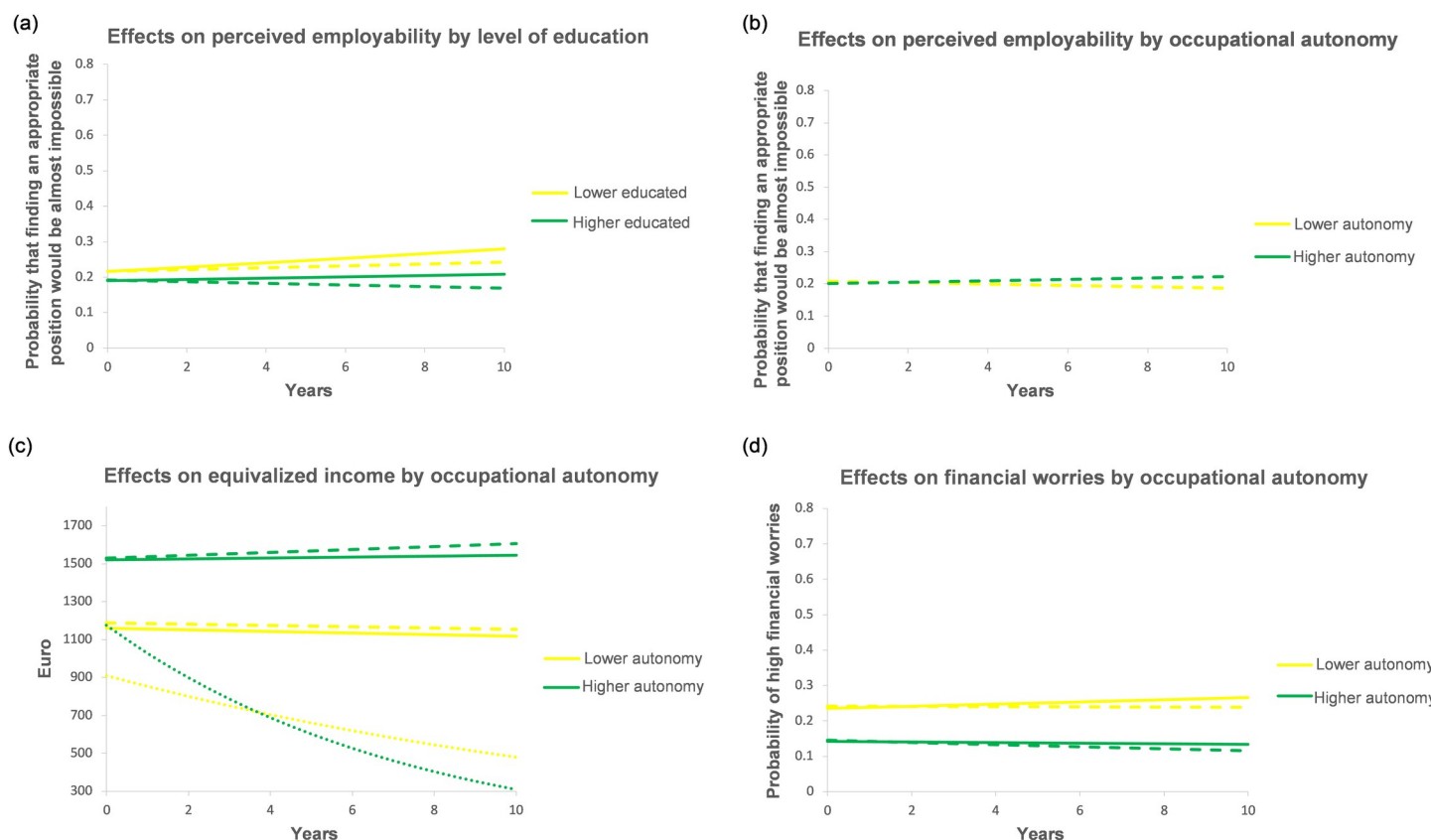

**Fig 5. Moderation of the within-level effects of employment status duration by person-specific mean SES across observations.** Dashed lines = effects of employment duration. Solid lines = effects of organizational tenure. Dotted lines = effects of unemployment duration.

Hypothesis 4b, which predicted that high SES would buffer against resource deterioration during continuous unemployment, was not supported. Educational attainment did not moderate the effects of unemployment duration on resources, whereas one significant moderating effect was found for occupational autonomy (Table 3). However, it was in the opposite direction than expected: Workers with higher average occupational autonomy across observations (when they had been employed) reported higher equivalized disposable income in the beginning of unemployment, but they experienced a steeper decrease in income with longer unemployment (from estimated 1,175 to 308 Euro after 10 years) than their counterparts with lower occupational autonomy (from 911 to 480 Euro; Fig 5C).

## Within-level effects of resources on SWB (second stage of mediation)

As no duration variable had significant effects on loneliness at the within level, I did not enter loneliness as a predictor at the second stage of mediation. Table 4 summarizes the within-level effects of six remaining resources on SWB in fully adjusted models (see full results in S3 Table). The within-level effects of resources reflected in how far scoring higher (or lower) than one's individual average on a particular resource at a given measurement occasion was related to higher (or lower) SWB at the same occasion. Economic resources had very small effects on SWB, some of them were not significant or not in the expected direction (for perceived employability). As regards social resources, occasion-specific frequency of socializing has significantly positive but likewise extremely small effects on concurrent SWB. In contrast,

**Table 4. Within-level effects of resources on SWB (second stage of mediation).**

| DVs and predictors | *B* (*SE*) | 95% CI | *p* | β |
|---|---|---|---|---|
| *DV*: *life satisfaction* | | | | |
| Income | 0.229 | [0.200, 0.258] | < .001 | .053 |
| Financial worries | -0.076 | [-0.095, -0.056] | < .001 | -.031 |
| Perceived employability | -0.012 | [-0.028, 0.003] | .114 | -.005 |
| Frequency of socializing | 0.006 | [0.004, 0.008] | < .001 | .020 |
| Social support availability | 0.382 | [0.299, 0.465] | < .001 | .292 |
| Mastery | 0.486 | [0.458, 0.513] | < .001 | .372 |
| *DV*: *emotional well-being* | | | | |
| Income | -0.002 | [-0.074, 0.070] | .961 | .000 |
| Financial worries | -0.116 | [-0.164, -0.069] | < .001 | -.053 |
| Perceived employability | -0.065 | [-0.102, -0.028] | < .001 | -.027 |
| Frequency of socializing | 0.010 | [0.005, 0.015] | < .001 | .035 |
| Social support availability | 0.322 | [0.146, 0.498] | < .001 | .271 |
| Mastery | 0.522 | [0.438, 0.606] | < .001 | .439 |

DV = dependent variable. Mastery and emotional well-being were modeled as latent variables. All effects are adjusted for the full set of control variables.

occasion-specific social support availability had sizable effects on both indicators of concurrent SWB, and mastery had even stronger effects.

## Within-level effects of employment status duration on SWB via resources

Table 5 shows main findings from the mediation model with multiple mediators, whereby Bayesian estimation was employed and categorical mediators were treated as latent continuous variables [75]. For these reasons, some path estimates deviated slightly from the previously reported MLR estimates for continuous outcomes. A longer unemployment duration significantly reduced life satisfaction via deterioration in economic (mainly lower income and higher financial worries) and social (mainly lower social support availability) resources. The total reduction in life satisfaction (Fig 2D) was small: 0.27 *SD* (at the within level) per 10 years of continuous unemployment, which stood in contrast to the large negative effect of unemployment itself (0.74 *SD*). Unemployment duration has no significant direct, indirect, or total effect on change in emotional well-being, whereas unemployment as such had a sizable negative effect of 0.36 *SD*. Thus, Hypothesis 5 was partly supported for life satisfaction and not supported for emotional well-being.

Furthermore, a longer duration of continuous employment had significant but extremely small positive effects on both SWB indicators via growth in economic resources (mainly lower financial worries), which corresponded to an improvement of less than 0.01 within-level *SD* of life satisfaction or emotional well-being per 10 years of continuous employment. The total effect (Fig 2D) amounted to 0.03 *SD* within the same period (not significant for emotional well-being). Thus, for employment duration, Hypothesis 5 was supported but only for mediation via economic resources.

Finally, a longer organizational tenure had significant but extremely small negative effects on both SWB indicators via decreasing mastery, which corresponded to a reduction of 0.02 *SD* (at the within level) of life satisfaction or emotional well-being per 10 years of tenure. The total effect (Fig 2D) was of about the same size and significant only for life satisfaction. Thus, for organizational tenure, Hypothesis 5 was supported for mediation via mastery, but this indirect effect and the total effect on SWB were unexpectedly negative.

**Table 5. Summary of direct, indirect, and total within-level effects of employment status duration on SWB.**

| Predictors and effects | Life satisfaction | | Emotional well-being | |
|---|---|---|---|---|
| | B | 95% CI | B | 95% CI |
| *Unemployment duration* | | | | |
| Direct | 0.023 | [-0.028, 0.079] | 0.045 | [-0.034, 0.125] |
| Via economic resources | -0.010 ** | [-0.014, -0.005] | -0.005 | [-0.011, 0.001] |
| Via social resources | -0.021 * | [-0.053, -0.002] | -0.018 | [-0.048, 0.001] |
| Via personal resources | -0.026 | [-0.068, 0.014] | -0.017 | [-0.053, 0.016] |
| Total | -0.035 * | [-0.064, -0.007] | 0.004 | [-0.067, 0.073] |
| *Employment duration* | | | | |
| Direct | 0.003 | [0.000, 0.007] | 0.004 | [-0.002, 0.010,] |
| Via economic resources | 0.001 * | [0.000, 0.001] | 0.001 * | [0.000, 0.001] |
| Via social resources | 0.000 | [-0.001, 0.002] | 0.001 | [-0.001, 0.002] |
| Via personal resources | 0.000 | [-0.003, 0.002] | 0.000 | [-0.003, 0.002] |
| Total | 0.004 ** | [0.002, 0.006] | 0.004 | [-0.001, 0.010] |
| *Organizational tenure* | | | | |
| Direct | 0.001 | [-0.002, 0.005] | -0.001 | [-0.008, 0.006] |
| Via economic resources | 0.000 | [0.000, 0.001] | 0.000 | [0.000, 0.001] |
| Via social resources | -0.001 | [-0.003, 0.000] | -0.001 | [-0.003, 0.001] |
| Via personal resources | -0.003 * | [-0.006, 0.000] | -0.002 * | [-0.005, 0.000] |
| Total | -0.003 ** | [-0.005, -0.001] | -0.004 | [-0.010, 0.002] |

Mastery and emotional well-being were modeled as latent variables. Via economic resources = sum of indirect effects via income, financial worries, and perceived employability. Via social resources = sum of indirect effects via frequency of socializing and social support availability. Via personal resources = indirect effect via the latent mastery variable. All effects were adjusted for the full set of control variables.

* $p < .05$.

** $p < .01$.

## Moderator analyses of the total effects on SWB

There was no significant (at $p < .01$) moderation of the effects of employment status duration on life satisfaction and emotional well-being by age, gender, educational attainment, and occupational autonomy (Table 6). Thus, the moderating effects on the first stage of mediation were not translated into any differences in the effects on SWB.

## Between-level effects

Between-level associations, which may point at selection effects, are shown in S2 and S3 Tables. Individuals who were unemployed at all occasions significantly differed from other participants only in their lower equivalized disposable income. A larger number of employment spells during the observation period (i.e., a more discontinuous employment history) was significantly associated with lower income, more financial worries, and lower life satisfaction, but also with higher perceived employability across observations. A larger number of unemployment spells was significantly associated with lower economic resources, lower frequency of socializing, lower mastery, and lower life satisfaction across observations. All effects were small or very small.

Furthermore, average duration of both continuous employment and organizational tenure was significantly associated with slightly higher average income (a 15 Euro difference per 10 years of duration) and negligibly higher frequency of socializing (a 0.02–0.04 between-level *SD* difference per 10 years of duration). Additionally, individuals with longer average

**Table 6. Moderation of the total within-level effects of employment status duration on SWB by between-level predictors.**

| Random effects (between level) | Life satisfaction | | Emotional well-being | |
|---|---|---|---|---|
| | *B* | 95% CI | *B* | 95% CI |
| **Average effect of employment duration (S1)** | 0.007 ** | [0.004, 0.010] | 0.002 | [-0.007, 0.010] |
| **S1 on average age** | 0.000 | [0.000, 0.000] | 0.000 | [-0.001, 0.000] |
| **S1 on sex (female)** | 0.000 | [-0.004, 0.003] | 0.009 * | [0.000, 0.017] |
| **S1 on average years of education** | 0.002 * | [0.001, 0.002] | 0.000 | [-0.002, 0.002] |
| **S1 on average occupational autonomy** | -0.001 | [-0.003, 0.001] | 0.000 | [-0.005, 0.006] |
| **Average effect of organizational tenure (S2)** | -0.004 * | [-0.008, -0.001] | -0.008 | [-0.017, 0.001] |
| **S2 on average age** | 0.000 | [0.000, 0.000] | 0.000 | [-0.001, 0.000] |
| **S2 on sex (female)** | 0.000 | [-0.003, 0.004] | 0.008 | [-0.002, 0.018] |
| **S2 on average years of education** | 0.001 | [0.000, 0.002] | -0.001 | [-0.004, 0.001] |
| **S2 on average occupational autonomy** | 0.000 | [-0.003, 0.002] | 0.001 | [-0.006, 0.008] |
| **Average effect of unemployment duration (S3)** | -0.152 * | [-0.300, -0.005] | 0.061 | [-0.146, 0.302] |
| **S3 on average age** | 0.002 | [-0.002, 0.007] | -0.004 | [-0.013, 0.003] |
| **S3 on sex (female)** | -0.017 | [-0.109, 0.080] | -0.041 | [-0.213, 0.102] |
| **S3 on average years of education** | -0.012 | [-0.036, 0.015] | 0.026 | [-0.028, 0.073] |
| **S3 on average occupational autonomy** | -0.030 | [-0.111, 0.046] | -0.006 | [-0.114, 0.122] |
| **Residual variance ($\sigma^2$) S1** | 0.003 ** | [0.002, 0.003] | 0.002 ** | [0.001, 0.003] |
| **Residual variance ($\sigma^2$) S2** | 0.003 ** | [0.003, 0.003] | 0.003 ** | [0.002, 0.004] |
| **Residual variance ($\sigma^2$) S3** | 0.194 ** | [0.140, 0.267] | 0.107 ** | [0.042, 0.239] |

CI = Bayesian credibility intervals. All moderators were entered in the same equation. Random S1 and S3 were estimated in the same model, random S2 was estimated in a separate model. Missing values on the predictor (employment status duration) could not be estimated in these models. For analyses with random S1 and S3, $N_{persons}$ = 28,911, $N_{observations}$ = 211,480. For analyses with random S2, $N_{persons}$ = 30,045, $N_{observations}$ = 223,312. The models were adjusted for the full set of control variables.

\* $p < .05$.

\*\* $p < .01$.

organizational tenure reported slightly lower financial worries (0.8 pp difference per 10 years), lower perceived employability (5.8 pp difference per 10 years) and slightly lower loneliness (0.3 pp difference per 10 years). Effects were larger for the average duration of continuous unemployment, which was significantly associated with lower income (a 439.7 Euro difference per 10 years), lower perceived employability (a 22.3 pp difference per 10 years), lower frequency of socializing (a 0.32 *SD* difference per 10 years), and lower mastery (a 0.56 *SD* difference per 10 years) across observations.

Finally, individuals with greater economic (excepting perceived employability), social, and personal resources also reported higher SWB than their less resourceful counterparts. With resources controlled for, employment status duration had very small and inconsistent associations with SWB at the between level.

## Discussion

In this study, I drew on cumulative (dis)advantage and conservation of resources theories [30, 32–36] to investigate whether stable employment (in terms of continuous employment or organizational tenure) is associated with the accumulation of economic, social, and personal resources, whereas continuous unemployment is associated with their depletion, which, in turn, may lead to positive or negative SWB change. Additionally, I tested whether such cumulative effects are amplified by timing (age), gender, and SES differences. I used large-scale

panel data from Germany (SOEP 1985–2012) and applied multilevel modeling with multiple observations nested within participants. In the following, I discuss key lessons learned.

## Lesson 1. Limited evidence for a spread of (dis)advantage beyond economic resources

In line with findings from earlier research [2–4, 8], individuals were in many respects worse off during unemployment occasions as compared with employment occasions: They reported substantially lower economic resources (lower equivalized disposable income, higher financial worries, and lower perceived employability), slightly lower social resources (only in terms of loneliness), and substantially lower personal resources (mastery) as well as lower SWB (especially life satisfaction but also emotional well-being). These differences supported both the economic and psychological deprivation perspectives on unemployment [5–7]. In the long term, remaining unemployed was associated with a further decrease in economic resources and with a qualitative deterioration of one's social support network (i.e., from feelings of loneliness to having no one to turn to in case of serious illness). Nevertheless, whereas unemployment itself was associated with disadvantage across multiple domains, the consequences of longer unemployment duration were mainly economic.

Life satisfaction also decreased during long-term unemployment, but the effect of unemployment duration, which was mediated by resource depletion, was much smaller than the average effect of unemployment as such. There were no significant effects of unemployment duration on emotional well-being. These findings concurred with previous research on SWB changes in response to life events [2, 42–45]. On the one hand, SWB is seen as highly adaptable, and emotional well-being more so than cognitive well-being (i.e., life satisfaction). This explains why reductions in SWB during long-term unemployment remain modest even as the depletion of important resources continues. On the other hand, unemployment was shown to alter the set point for SWB even after re-employment [43]. Thus, it is no wonder that workers consistently reported lower SWB during unemployment than they did on employment occasions.

Furthermore, although most effects of continuous employment and organizational tenure were also restricted to the economic domain, several effects on social and personal resources did emerge (see next section). These findings partly support the concepts of accumulation of (dis)advantage across domains (CAD; [32]) or resource caravans (COR; [35, 36]). However, most effects of continuous employment and organizational tenure were extremely small. Besides, the direction of accumulation was not always the same across domains. Possibly for these reasons, the effects of continuous employment and organizational tenure on SWB were negligibly small, even though their mediation by resources was also supported.

## Lesson 2. Younger workers and women benefit from remaining employed more

The CAD theory has traditionally assigned importance to the timing of (dis)advantage, whereby earlier timing is supposed to lead to larger consequences [32, 34]. However, in my study, few differences between relatively younger and relatively older workers in the effects of employment status duration emerged. The only significant difference indicated that with a longer duration of continuous employment or organizational tenure, perceived employability of younger workers improved, whereas that of older workers substantially decreased. Older workers are often confronted with old-age stereotypes and discriminatory practices of employers [82]. Besides, in Germany, strong employment protection and relatively small wage differentials across employers lead to low external job mobility [38, 39]. Older workers in particular may have achieved a secure and well-paid position, but they have poor chances of being hired

elsewhere with equally good conditions. In contrast, during early career, when precarious working conditions and job changes are not unusual, more continuous employment or a longer organizational tenure may signal occupational success and thereby improve future employment chances.

Findings on the role of gender, a traditional dimension of inequality [30, 34], were also partly unexpected. Whereas few gender differences in the economic correlates of employment status duration emerged, women workers appeared to be better able to accumulate social (frequency of socializing) and personal (mastery) resources during continuous employment or their time with the firm. Their perceived employability also improved. Paradoxically, these beneficial effects might be attributed to the structural disadvantage of women in the German labor market (note that the SOEP started in the 1980s). Even in the 2010s, German women still had more family-related career interruptions and less ambitious careers than men did [40]. Consequently, those women who showed a more continuous career pattern might be special persons who highly efficiently managed their careers, or they might enjoy a special support from their work environments (cf. [36]), or they might have a comparative advantage in the labor market against the backdrop of their counterparts with more discontinuous careers. On the negative side, a higher competition and a pressure to perform might take a toll on personal resources (mastery) of men workers as they accumulated work experience [83].

## Lesson 3. The SES gap in economic but not in social and personal resources grows with a longer employment duration

With a longer duration of continuous employment or organizational tenure, the gap in equivalized disposable income and financial worries between the workers with higher and lower occupational autonomy slightly increased. In addition, a higher educational attainment buffered the negative effects of continuous employment and organizational tenure on perceived employability. These findings support the notion that economic inequality increases not only between employed and unemployed individuals but also, albeit to a lesser extent, between stably employed workers with differing SES (cf. [50, 60]). Although SES differentials in social and personal resources were also observed, they did not increase with more continuous employment experience. This negative finding raises the question whether the SES disparities in psychosocial resources remain generally stable over the life course or whether they increase with accumulation of some other experience than stable employment. Furthermore, contrary to expectations, a higher SES did not buffer against the negative effects of long-term unemployment (cf. [3, 4]). In the German context, the social safety net prevents unemployed individuals with all SES backgrounds from slipping into acute poverty, making a higher SES not necessarily an advantage in coping with unemployment.

## Lesson 4. Resource gain during employment is less pronounced than resource loss during unemployment

During prolonged unemployment, workers experienced sizable resource depletion and reductions in life satisfaction. However, the resource growth during continuous employment, if any, was incremental, just as were the benefits to life satisfaction. These findings support the views that advantage is not the polar opposite of disadvantage [57] and that resource accumulation is cumbersome [34, 36]. Continuous employment may mainly serve to preserve resources, even though some workers may be able to build them up. For instance, the modern version of the job demands–resources theory [60] asserts that workers who engage in job crafting (i.e., proactive changes of different aspects of their work) may experience gain spirals. In contrast, those who engage in self-undermining (i.e., behaviors that create obstacles and increase job

demands) or have high job demands and low job resources experience loss spirals [60]. More-over, continuously employed workers may accumulate career-specific resources, such as knowledge and skills, motivational, and environmental resources [61], rather than general resources addressed in the present study. Finally, organizational context also matters in enabling or hindering resource accumulation [24, 36].

## Lesson 5. Workers benefit from staying continuously employed rather than from staying with the same employer

A long organizational tenure remains very common in Germany. All the more instructive is therefore my finding that it might have both positive (i.e., for income and frequency of social-izing) and negative (i.e., for perceived employability and mastery) consequences for resources and, on balance, negative effects on SWB. On the positive side, in Germany, staying with the same firm often brings higher job security and salary increases. As longer-tenured workers are less afraid to lose their jobs and have developed an extensive social network in the company, they may find more time and opportunities for socializing than newcomers. On the negative side, staying with the same firm may limit the opportunities for varied experiences and profes-sional growth (cf. job plateaus; [84]), leading the workers to doubt their value on the labor market and their ability to exercise initiative at work (cf. [85]). To some workers, staying with the same employer may be due to "nonevents" such as not receiving a better offer elsewhere [86], which may account for my finding on the negative consequences for mastery. Although the effects should not be overinterpreted because of their very small sizes, it appears that stay-ing continuously employed but changing employers has more benefits.

## Lesson 6. More evidence for self-selection into (long-term) unemployment than into stable employment

As between-level findings indicated, individuals with lower average economic, social, and per-sonal resources across observations became unemployed somewhat more often and experi-enced longer unemployment spells during the observation period. Average SWB was hardly associated with one's unemployment history across observations. Still, these findings corrobo-rated earlier evidence on the self-selection of individuals with low resources and poor mental health into (long-term) unemployment [15, 19, 22].

For the number of employment spells, average duration of continuous employment, and average organizational tenure, several significant but negligibly small associations with average economic and social resources and SWB across observations emerged. Most of these associa-tions went into the expected direction (i.e., the higher the resources and SWB, the more con-tinuous one's employment history). The negligibly small size of these associations may have very different explanations. First, highly stable and continuous employment histories (com-pared to trajectories interrupted by normative events such as parenthood) may have more to do with (chance) life circumstances than with workers' resources and mental health. Second, stable employment might have been so common in Germany in the observation period that individual differences played little role in achieving it. Third, resourceful German workers may not take pains to remain continuously employed or with the same employer exactly because they perceive little benefits from such employment stability.

## Limitations and future directions

This study was based on secondary data analysis, with corresponding limitations. I had to rely on suitable instruments that were available in the data. The validity of some measures may be

questioned; for instance, perceived employability in employed individuals might have a negative connotation (e.g., easiness of finding a comparable position might imply that the current job is not that attractive). Some constructs were not assessed regularly, and alternating versions of items and rating scales were sometimes used. Hence, I had to utilize fewer items per construct, which were more similar across waves, and applied ranking procedures to equalize alternating versions. Moreover, I calculated employment status duration from different sources of information, and certain gaps and inconsistencies in the data remained despite many counterchecks. For the main-effects analyses, I checked whether my findings held if employment and unemployment duration were coded somewhat differently, namely, if continuous employment allowed for no more than one-month interruptions, whereas unemployment duration was based on registered unemployment (instead of ILO criteria). Results for employment duration were essentially the same as reported above; additionally, a very small positive within-level effect on mastery emerged. Duration of registered unemployment had no significant effects on financial worries, smaller effects on perceived employability and social support availability, and no significant total effect on life satisfaction.

The SOEP is the largest and the most representative panel study in Germany. Still, it is affected by respondent attrition. Even though I applied full information estimators and included missing data covariates, attrition might influence the present findings. For example, the effects of long-term unemployment might be underestimated if those unemployed individuals who experienced substantial mental health deterioration systematically dropped out of the panel. Furthermore, I tested a mediation model at the within-person level, which made certain assumptions about the direction of effects. The predictors (employment status duration), the mediators (resources), and the outcomes (SWB) were occasion-specific deviations from individual averages. Whereas employment status duration referred to the time up until a given measurement occasion, the association between occasion-specific resources and SWB represented covariation in time. Thus, another direction of effects—from SWB to some or all of the resources considered—would be also compatible with the data. Most likely, a bidirectional influence exists between the ups and downs in resources and SWB.

Potential cohort differences or historical shifts in the meaning of employment status and its duration were not considered in this study. Despite the impressive overall observation period of 28 years, many mediator variables were assessed only several times, making period/cohort analyses with mediators infeasible. Moreover, very long-term unemployment has become increasingly rare after the Hartz reform of the mid-2000s [38]. For the sake of comparability between employment and unemployment duration, I reported effect sizes for all mediators and outcomes for up to 10 years of duration. The majority of long-term unemployed individuals experienced far shorter unemployment spells, but there were in fact 1–12 plausible records in almost every wave (except for 2008) with unemployment duration of 10+ years.

Future studies may further investigate interindividual variability in the trajectories of advantage and disadvantage associated with paid employment. Important sources of such variability could be organizational context, type of employment (i.e., full-time, part-time, or marginal), job stress, P–E fit, and agency. At the micro level, researchers may use intensive longitudinal designs to address the aspects of work environments, workers' characteristics, experiences, and behaviors that foster preservation, accumulation, or depletion of general and career-specific resources. At the macro level, the consequences of long-term employment or unemployment may be compared across different regional and country contexts. For instance, in more liberal and flexible labor market regimes, long-term unemployment is even less frequent but may lead to more economic and mental health problems than it is the case in Germany. Similarly, long tenure is less normative in such countries and may have even less beneficial effects on workers' psychosocial resources and SWB. Even within one country,

varying regional economic conditions may lead to unemployment being more or less normative and employment being more or less rewarding. Future research may also attempt to distinguish between a "fair" and "unfair" growth in inequality between workers. The former can be traced back to the differences in ability and effort and may signal to workers that their exertions (e.g., labor force attachment and work effort) make a difference. In contrast, the latter is a true CAD process, which is blind to individual merit [30, 33].

## Conclusions

From the early 1980s to the 2010s, German workers who stayed continuously employed did not seem to accumulate—to any substantial degree—economic, social, or personal resources over time. However, continuous employment might have helped them to conserve resources, especially economic ones, and thereby to maintain SWB. Workers who experienced employment interruptions but stayed with the same employer benefited less, but overall, they also exhibited a high stability in resources and SWB. Major sociodemographic dimensions of inequality (age, gender, and SES) made a little difference in this picture of long-term stability, but still, younger workers, women, and higher-SES workers appeared to derive more benefits from continuous employment. In contrast, long-term unemployed individuals experienced depletion of mainly economic but also social resources, which contributed to a further deterioration in life satisfaction. Even so, these unfavorable effects were lesser in magnitude than the stark negative effects of the initial unemployment event. In general, the highly regulated German labor market and social security system may both dampen the rewards of a strong labor force attachment and buffer against the losses of long-term unemployment. Still, my findings may be generalizable to other Western European countries with comparable labor market regimes (e.g., Austria, Belgium, or the Netherlands). A policy recommendation for such countries may be to loosen their employment regulations in a direction that would lead to more employment (and job change) opportunities to workers of different age and would better reward continuous employment.

## Supporting information

**S1 Table. Bivariate correlations between the study variables at the between and within levels.**
(PDF)

**S2 Table. Multilevel regression results for the main effects of employment status duration on resources (first stage of mediation).**
(PDF)

**S3 Table. Multilevel regression results for the effects of employment status duration and resources on SWB (second stage of mediation).**
(PDF)

## Acknowledgments

The data from the German Socio-Economic Panel were made available to the author by the DIW Berlin via a data distribution contract.

## Author Contributions

**Conceptualization:** Maria K. Pavlova.

**Formal analysis:** Maria K. Pavlova.

**Investigation:** Maria K. Pavlova.

**Methodology:** Maria K. Pavlova.

**Writing – original draft:** Maria K. Pavlova.

**Writing – review & editing:** Maria K. Pavlova.

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
