## [Decision Letter · Decision Letter 0]

1 Dec 2021

PONE-D-21-35645Do Workers Accumulate Resources During Continuous Employment and Lose Them During Unemployment, and What Does That Mean for Their Subjective Well-Being?PLOS ONE

Dear Dr. Pavlova,

Thank you for submitting your manuscript to PLOS ONE. After careful consideration, we feel that it has merit but does not fully meet PLOS ONE’s publication criteria as it currently stands. Therefore, we invite you to submit a revised version of the manuscript that addresses the points raised during the review process. You should address/respond all comments but there is no need to shorten the paper as stated in one of the reports.

We look forward to receiving your revised manuscript.

Kind regards,

Petri Böckerman

Academic Editor

PLOS ONE

Journal Requirements:

Reviewers' comments:

Reviewer's Responses to Questions

**Comments to the Author**

1. Is the manuscript technically sound, and do the data support the conclusions?

Reviewer #1: Yes

Reviewer #2: Partly

2. Has the statistical analysis been performed appropriately and rigorously? 

Reviewer #1: Yes

Reviewer #2: Yes

3. Have the authors made all data underlying the findings in their manuscript fully available?

Reviewer #1: No

Reviewer #2: Yes

4. Is the manuscript presented in an intelligible fashion and written in standard English?

Reviewer #1: Yes

Reviewer #2: Yes

5. Review Comments to the Author

Reviewer #1: 1. The revised introduction should motivate the empirical context (i.e., focus on Germany).

2. Is attrition an issue in the data? Individuals with lowest level of health status may be much more likely to drop from the panel over time. Does this have implications for the interpretation of the results that are presented in the paper?

3. Is educational attainment predetermined control variable or not?

4. Is there any need to account for local economic conditions that may be relevant for both health and employment status?

5. There is earlier empirical literature on health and labor market status in health economics using panel data (https://doi.org/10.1002/hec.1361). This issue should be acknowledged in the revised version.

6. What are the key policy lessons for other countries? What is the external validity of the estimates?

Reviewer #2: The manuscript is judged to be meaningful as data analyzed using objective indicators in various fields. However, it is hoped that the manuscript will be improved through corrections in some parts.

1) Overall, it provides a lot of information and is considered to be useful, but I hope that it will be condensed.

2) In relation to well-being, age, gender, and SES were considered, but other variables were not considered. Why? For example, if an industrial accident occurs, the results of this study may differ, and this needs to be considered. Although such a study may be meaningful, it is judged that the results may be meaningless if the missing parameters are considered.

3) It would be good if you divide the manuscript into introduction, theory, method, result and discussion, and conclusions. In particular, there is a disadvantage that the introduction part is too long and the research result part looks weak.

6. PLOS authors have the option to publish the peer review history of their article (what does this mean?). If published, this will include your full peer review and any attached files.

Reviewer #1: **Yes: **Petri Böckerman

Reviewer #2: No

---

## [Author Response · Author response to Decision Letter 0]

6 Dec 2021

Journal Requirements: 

I have completely revised the manuscript to be in line with PLOS ONE style. Among other things, I converted footnotes into main text, which shows in the markup.

In the original submission, I specified: 

“The data underlying the results presented in the study are available from the DIW Berlin via a data distribution contract. The dataset doi: https://doi.org/10.5684/soep.v30.”

The SOEP data are owned by a third party (DIW Berlin) but are freely accessible to all researchers who register themselves as users. I may not share the data, not even my working dataset. Please let me know whether I should modify my Data Availability statement or do something else.

Done.

I have checked all references, there are no retracted or corrected articles in the list.

Reviewers' comments:

Reviewer #1: 1. The revised introduction should motivate the empirical context (i.e., focus on Germany).

In the revision, I moved the paragraph on the German context from “The Present Study” to the Introduction and added a sentence that explains why this context is interesting for my research questions (pp. 3-4).

2. Is attrition an issue in the data? Individuals with lowest level of health status may be much more likely to drop from the panel over time. Does this have implications for the interpretation of the results that are presented in the paper?

I used satisfaction with health from the previous measurement as one of missing value covariates (see p. 23). In addition, wherever possible (i.e., with maximum likelihood estimation but not with Bayesian estimation), I included the dependent variable from the previous wave as a missing value covariate. As most dependent variables referred to well-being, they can be seen as mental health indicators. In the revision, I added to Limitations (pp. 44-45):

“Even though I applied full information estimators and included missing data covariates, attrition might influence the present findings. For example, the effects of long-term unemployment might be underestimated if those unemployed individuals who experienced substantial mental health deterioration systematically dropped out of the panel.”

3. Is educational attainment predetermined control variable or not?

Educational attainment was always a control variable at both within and between levels, even in the models where average educational attainment served as a moderator at the between level. To make it clearer, I added in the revision (see p. 22):

“All time-varying control variables were assessed yearly. Most of them (except for the characteristics of the current spell, which were purely within-level variables) were also included as control variables at the between level (see S2 Table for a full list of control variables at both levels).”

4. Is there any need to account for local economic conditions that may be relevant for both health and employment status?

It is possible that there is regional variation in the effects. However, addressing it would go beyond the scope of the present study, which focused on the individual level (and is even so very extensive). In the revision, I added to future directions (p. 46):

“At the macro level, the consequences of long-term employment or unemployment may be compared across different regional and country contexts… Even within one country, varying regional economic conditions may lead to unemployment being more or less normative and employment being more or less rewarding.”

5. There is earlier empirical literature on health and labor market status in health economics using panel data (https://doi.org/10.1002/hec.1361). This issue should be acknowledged in the revised version.

Thank you for sharing this reference. It is highly relevant indeed, and I have cited it multiple times in the Introduction and Discussion.

6. What are the key policy lessons for other countries? What is the external validity of the estimates?

I added to Conclusions (p. 47): “In general, the highly regulated German labor market and social security system may both dampen the rewards of a strong labor force attachment and buffer against the losses of long-term unemployment. Still, my findings may be generalizable to other Western European countries with comparable labor market regimes (e.g., Austria, Belgium, or the Netherlands). A policy recommendation for such countries may be to loosen their employment regulations in a direction that would lead to more employment (and job change) opportunities to workers of different age and would better reward continuous employment.”

Reviewer #2: The manuscript is judged to be meaningful as data analyzed using objective indicators in various fields. However, it is hoped that the manuscript will be improved through corrections in some parts.

1) Overall, it provides a lot of information and is considered to be useful, but I hope that it will be condensed.

As suggested by the Academic Editor, I did not condense the manuscript. I should note that I did shorten the manuscript previously, but it remains so long because of the complexity of the research questions and data analyses and also because I wanted to avoid piecemeal publication of findings.

2) In relation to well-being, age, gender, and SES were considered, but other variables were not considered. Why? For example, if an industrial accident occurs, the results of this study may differ, and this needs to be considered. Although such a study may be meaningful, it is judged that the results may be meaningless if the missing parameters are considered.

This is very true, there may be multiple factors and events that might influence accumulation or depletion of resources and well-being. However, I considered an extensive set of control variables at both levels (see p. 22). Among other things, I controlled for health satisfaction and disability – both at the current measurement occasion at the within level and on average (or, in case of disability, on all occasions vs. only some) at the between level. I believe that the effects of an industrial accident would be reflected in a change in these variables, which are controlled for. As regards my choice of moderators (age, gender, SES), it was purely theoretically driven, because CAD theory attempts to explain inequality, whereas these variables are major factors of inequality.

3) It would be good if you divide the manuscript into introduction, theory, method, result and discussion, and conclusions. In particular, there is a disadvantage that the introduction part is too long and the research result part looks weak.

I corrected the revision to be in line with PLOS One style guidelines. The apparent weakness of the research part might have to do with tables and figures being absent from the manuscript body (in the original submission). In the revision, in line with PLOS style, tables are directly in the manuscript body and figure captions as well.

Additional changes

1. I spotted another useful reference and added it to the manuscript:

Stiglbauer B, Batinic B. The role of Jahoda’s latent and financial benefits for work involvement: A longitudinal study. J Vocat Behav. 2012;81: 259–268. doi:10.1016/j.jvb.2012.07.008

2. I changed figures from grayscale to color, because this is an online-only publication, and color makes figures better readable.

---

## [Decision Letter · Decision Letter 1]

10 Dec 2021

Do workers accumulate resources during continuous employment and lose them during unemployment, and what does that mean for their subjective well-being?

PONE-D-21-35645R1

Dear Dr. Pavlova,

We’re pleased to inform you that your manuscript has been judged scientifically suitable for publication and will be formally accepted for publication once it meets all outstanding technical requirements.

Kind regards,

Petri Böckerman

Academic Editor

PLOS ONE

Additional Editor Comments (optional):

Reviewers' comments:

Reviewer's Responses to Questions

**Comments to the Author**

1. If the authors have adequately addressed your comments raised in a previous round of review and you feel that this manuscript is now acceptable for publication, you may indicate that here to bypass the “Comments to the Author” section, enter your conflict of interest statement in the “Confidential to Editor” section, and submit your "Accept" recommendation.

Reviewer #1: All comments have been addressed

2. Is the manuscript technically sound, and do the data support the conclusions?

Reviewer #1: Yes

3. Has the statistical analysis been performed appropriately and rigorously? 

Reviewer #1: Yes

4. Have the authors made all data underlying the findings in their manuscript fully available?

Reviewer #1: No

5. Is the manuscript presented in an intelligible fashion and written in standard English?

Reviewer #1: Yes

6. Review Comments to the Author

Reviewer #1: I am happy with the revised version of the paper. I like the research question, the structure of the

paper, the quality of writing, and the way the authors describe their empirical proceeding and results. Most importantly, the authors have addressed all the issues stated in my referee report for the first version appropriately.

7. PLOS authors have the option to publish the peer review history of their article (what does this mean?). If published, this will include your full peer review and any attached files.

Reviewer #1: No

---

## [Editor Report · Acceptance letter]

13 Dec 2021

PONE-D-21-35645R1 

Do workers accumulate resources during continuous employment and lose them during unemployment, and what does that mean for their subjective well-being? 

Dear Dr. Pavlova:

I'm pleased to inform you that your manuscript has been deemed suitable for publication in PLOS ONE. Congratulations! Your manuscript is now with our production department. 

Kind regards, 

on behalf of

Professor Petri Böckerman 

Academic Editor

PLOS ONE